# Structural and functional analysis of LIM domain-dependent recruitment of paxillin to αvβ3 integrin-positive focal adhesions

Marta Ripamonti [1], Nicolas Liaudet [2], Latifeh Azizi[3], Daniel Bouvard[4], Vesa P. Hytönen [3,5] & Bernhard Wehrle-Haller [1✉]

The LIM domain-dependent localization of the adapter protein paxillin to β3 integrin-positive focal adhesions (FAs) is not mechanistically understood. Here, by combining molecular biology, photoactivation and FA-isolation experiments, we demonstrate specific contributions of each LIM domain of paxillin and reveal multiple paxillin interactions in adhesion-complexes. Mutation of β3 integrin at a putative paxillin binding site ($\beta3^{VE/YA}$) leads to rapidly inward-sliding FAs, correlating with actin retrograde flow and enhanced paxillin dissociation kinetics. Induced mechanical coupling of paxillin to $\beta3^{VE/YA}$ integrin arrests the FA-sliding, thereby disclosing an essential structural function of paxillin for the maturation of β3 integrin/talin clusters. Moreover, bimolecular fluorescence complementation unveils the spatial orientation of the paxillin LIM-array, juxtaposing the positive LIM4 to the plasma membrane and the β3 integrin-tail, while in vitro binding assays point to LIM1 and/or LIM2 interaction with talin-head domain. These data provide structural insights into the molecular organization of β3 integrin-FAs.

[1] Department of Cell Physiology and Metabolism, University of Geneva, Centre Médical Universitaire, Geneva 4, Switzerland. [2] Bioimaging Core Facility, Faculty of Medicine, University of Geneva, Geneva 4, Switzerland. [3] Faculty of Medicine and Health Technology, Tampere University, Tampere, Finland. [4] Montpellier Cell Biology Research Center (CRBM), University of Montpellier, CNRS UMR 5237, Montpellier, France. [5] Fimlab Laboratories, Tampere, Finland. ✉email: Bernhard.Wehrle-Haller@unige.ch

The anchoring of cells to the extracellular matrix (ECM) is a critical function of integrins. Yet, these heterodimeric receptors are also biomechanical sensors that trigger different signaling pathways in response to extracellular stimuli[1,2]. To accomplish these mechanical and signaling functions, upon ligand binding, integrins cluster in the plasma membrane (PM) and recruit the cytoplasmic adapter proteins talin and kindlin[1,3–5]. Following the mechanical coupling to the actin cytoskeleton, notably through C-terminal domain of talin, signaling adapter proteins, such as paxillin and FAK, are recruited to form nascent adhesions[6–8]. Further maturation of these complexes into larger focal adhesions (FAs) occurs in response to additional tension[9] and recruitment of reinforcing adapter proteins, such as vinculin, that further cross-link talin to the actin cytoskeleton[10,11].

The N-terminal domain of paxillin contains a proline-rich SH3-binding[12] and five LD motifs. Each of them, acting as a discrete protein-binding domain, is responsible for the interaction with structural proteins, such as vinculin, talin, and the ILK/Pinch/parvin complex[10,13–16]. In addition, the LD motifs bind signaling components such as kinases (e.g. FAK) and small GTPases which control the polymerization of the actin cytoskeleton[17,18]. The activity of the LD motifs, and notably the association with FAK and vinculin, is controlled by numerous phosphorylation sites (Tyr, Ser, Thr), which make paxillin an essential hub in the assembly of FAs and a pivotal player in coordinating integrin-downstream signaling pathways[10,19,20].

On the other hand, the paxillin C-terminus is composed of four highly similar LIM domains, that constitute the FA-targeting motif[21]. Each LIM domain, consisting of two contiguous zinc-fingers stacked together through a shared hydrophobic core[22,23], mediates protein–protein interactions, to create scaffolds for the assembly of multiprotein complexes at defined cellular sites[24–29]. Structural information of individual LIM domains revealed specific sites of interactions, as for example between the ankyrin repeats of ILK and the first LIM domain of Pinch[23]. Although the biochemical and structural nature of most of the proposed LIM domain-mediated interactions is not yet defined precisely, many proteins containing multiple LIM domains are recruited to FAs under mechanical tension[30–33]. It was therefore postulated that LIM domains could function as tension sensors[32] and proposed to act as localizers, targeting proteins to specific subcellular locations, such as tensioned or injured F-actin networks[22,32,34,35]. In the case of paxillin, a LIM3-, and partially LIM2-, dependent FA localization was proposed based on truncations and site-specific mutagenesis experiments[21]. Additional work in paxillin- and Hic-5-null embryonic stem cells further confirmed this notion, however, proposing that LIM1 and LIM4 only marginally contribute to the FA-targeting[36]. Recently, kindlin-dependent mechanisms of paxillin recruitment to FAs were proposed[37–39]. However, in attached talin-deficient cells, paxillin remained diffuse in the cytoplasm[40], suggesting that kindlin may not be the only determinant for paxillin recruitment to FAs. Some reports have shown a direct binding of LIM domain-containing proteins, such as the four and a half LIM domain protein 2 (FHL-2), to β integrin cytoplasmic tails[41]. In addition, a point mutation in the β1 integrin tail, which mimics the high-affinity talin-binding site naturally present in the alternatively spliced β1D integrin, negatively affected the localization of paxillin to adhesions[42], proposing that paxillin recruitment is also directly controlled by the β integrin cytoplasmic tail. This report is consistent with data demonstrating a direct role for the β3 integrin Tyr[747] residue, located in the membrane-proximal talin-binding NPLY[747] motif, in controlling paxillin localization to β3 integrin complexes[43,44].

Interestingly, modifications of this sequence or nearby residues, affected paxillin binding to adhesions[44,45]. Importantly, even in the presence of functional integrin/talin complexes that induced integrin activation and clustering, paxillin recruitment still required their mechanical linkage to the actin cytoskeleton[46,47]. Remarkably, paxillin displayed a mechanosensitive behavior on stress fibers[35] and could bind detergent extracted and stretched cytoskeletons, proposing the existence of tension-induced paxillin-binding sites[30]. Combined with the notion that some LIM domains can bind Tyr-containing motifs[48,49] similar to the NPLY sequence of β3 integrin, we previously hypothesized that paxillin LIM domains recognize the β3 integrin Y[747]-presenting motif in a tension-dependent manner[43].

While the central role of paxillin in the signal transduction from the integrin/talin/kindlin complex is well accepted, it is still controversial how it interacts with these proteins and whether it directly binds β integrin subunits[16,21,50–52]. Aiming for a better understanding of the integrin-dependent mechanisms of paxillin recruitment to β3 integrin-containing FAs, we studied the previously reported paxillin-binding defective β3[VE/YA] integrin in more detail[44]. This integrin, which contains the VE high-affinity talin-binding motif from layilin combined with the Y[747]A mutation, affected β3 integrin-dependent spreading on vitronectin and reduced the recruitment of endogenous paxillin to FAs[44]. Here we show that β3[VE/YA] integrin-containing adhesions exhibit remarkable stability defects, despite almost normal dissociation dynamics of talin and β3 integrin. The mechanical coupling of paxillin to such an integrin or the PM, rescued the instability of β3[VE/YA] integrin-FAs, highlighting a critical role of paxillin in the mechanical stabilization of integrin/talin complexes, possibly mediating the inside-out activation of the receptor and its binding to extracellular ligands. By elucidating the mechanisms of paxillin recruitment to αvβ3 integrin-containing adhesions, we revealed a defined LIM domain-dependent spatial orientation of paxillin, in which LIM4 directs towards the PM. We also demonstrated that upon deletion of LIM3, the strong paxillin-binding defect is caused by a steric misplacement of LIM1 and LIM2, forming a major binding element to β3 integrin adhesions. This proposes a new model in which the proper spacing of the LIM domains, as well as their correct positioning, is critical for reading the presence of mechanical tension occurring in the β3 integrin/talin/kindlin/F-actin linkage.

## Results

**Development of a cellular system to study paxillin recruitment and binding to αvβ3 integrin-positive adhesions.** The analysis of paxillin recruitment and binding to αvβ3 integrin-containing FAs was achieved by expressing recombinant β3 integrins in a previously characterized clone of NIH-3T3 fibroblasts which exhibits very low levels of endogenous β3 integrin[44]. Because of the ability of the β3 integrin subunit to only combine with either αv or αIIb, and because the αIIb chain is not expressed in NIH-3T3 fibroblasts, exogenous β3 integrins only dimerize with endogenous αv, therefore forming the αvβ3 integrin receptor that we sought to study. To better understand the mechanism of paxillin recruitment to β3 integrin-containing FAs, we employed the previously reported β3 integrin mutants, β3[VE] and β3[VE/YA] (Fig. 1a), known to enhance the binding of talin, and disrupting the potential paxillin-binding site[44]. While the first mutant presents the high-affinity talin-binding motif (VE), the second additionally contains the Y[745]A mutation, which created a spreading-incompetent integrin, causing reduced co-localization of endogenous paxillin[44]. Although the minimal levels of endogenous β3 integrin previously appeared unable to circumvent the

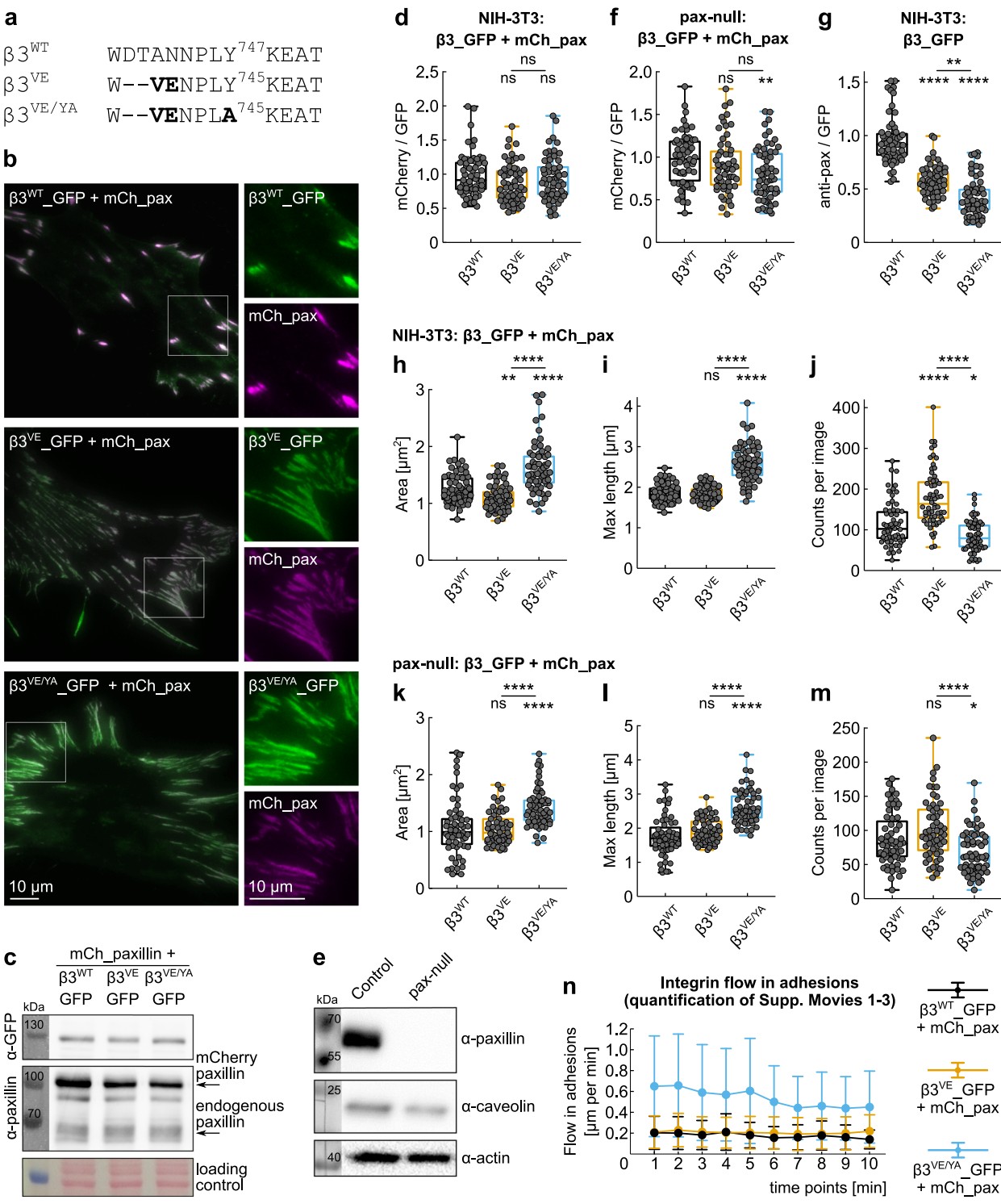

**Fig. 1**

defective signaling of transfected β3 integrin mutants in β3 integrin-dependent spreading assays performed in serum-free conditions and on a vitronectin-coated surface[44], prolonged culturing of β3VE/YA integrin-expressing NIH-3T3 fibroblasts in the presence of serum allowed spreading, FA assembly and their phenotypic analysis (Fig. 1b).

The three integrin proteins selected for this study (Fig. 1a), as well as mCherry_paxillin could be transiently expressed at comparable levels in NIH-3T3 fibroblasts (Fig. 1b, c and Supplementary Fig. 1a). Interestingly, the overexpressed mCherry_paxillin localized equally well to FAs, independently of the

type of β3 integrin subunit expressed as a fusion with GFP, as demonstrated by the quantification of its mCherry fluorescence intensity in respect to the GFP signal (Fig. 1d). In paxillin-deficient fibroblasts (Fig. 1e and Supplementary Fig. 1b), a very similar result was obtained (Fig. 1f). Nevertheless, a small but significant decrease of the paxillin/β3 integrin ratio was observed in β3VE/YA integrin-containing FAs (Fig. 1f), proposing that the overexpressed paxillin could mask a recruitment defect to this integrin. This notion was confirmed when the recruitment of endogenous paxillin, in the absence of overexpressed mCherry_paxillin, was measured in respect to the transiently transfected β3

**Fig. 1 Characterization of wild-type and mutant β3_GFP integrin-positive adhesions. a** Sequence alignment of β3 integrins, wt, and mutants, at the NPLY[747] motif. The high-affinity talin-binding chimeric β3 integrins were obtained by introducing the VE motif from layilin. Bold indicates mutated residues. **b, c** NIH-3T3 cells transiently transfected with β3_GFP integrins, wild-type or mutants, and mCherry_paxillin. **b** Representative TIRF images of cells fixed 24 h after spreading on glass coverslips in serum-containing medium and (**c**) recombinant protein expression assessed by western blotting against GFP and total paxillin. Loading control: Ponceau staining. **d, f** Quantification of paxillin within β3 integrin-positive adhesions, expressed as red/green ratio measured upon co-expression of mCherry_paxillin and GFP-tagged integrins in **d** NIH-3T3 and **f** paxillin-null fibroblasts. Statistical analysis is provided in Supplementary Dataset 1. **e** WB of the paxillin-null clone and the parental cells, showing the loss of paxillin expression. **g** Quantification of endogenous paxillin within β3 integrin-positive adhesions, expressed as red/green ratio measured upon expression of GFP-tagged integrins and staining with the α-paxillin mAb. Statistical analysis is provided in Supplementary Dataset 1. **h, i** Quantification of the (**h**) area and (**i**) maximal length of β3 integrin-positive adhesions in NIH-3T3 cells co-transfected with mCherry_paxillin and β3_GFP integrins, expressed as mean value per acquisition. Statistical analysis is provided in Supplementary Dataset 1. **j** Number of β3 integrin-positive adhesions per acquisition in NIH-3T3 cells co-transfected with mCherry_paxillin and β3_GFP integrins. Statistical analysis is provided in Supplementary Dataset 1. **k, l** Quantification of the **k** area and **l** maximal length of β3 integrin-positive adhesions in paxillin-null cells co-transfected with mCherry_paxillin and β3_GFP integrins, expressed as mean value per acquisition. Statistical analysis is provided in Supplementary Dataset 1. **m** Number of β3 integrin-positive adhesions per acquisition in paxillin-null cells co-transfected with mCherry_paxillin and β3_GFP integrins. Statistical analysis is provided in Supplementary Dataset 1. **n** Quantification of the β3 integrin flow in adhesions for the Movies 1, 2, and 3. Data points represent the mean of the optical flow of all pixels analyzed (i.e. within adhesions) between two consecutive frames; error bar ±SD. ns (not significant) $p > 0.05$; *$p \leq 0.05$; **$p < 0.01$; ***$p < 0.001$; ****$p < 0.0001$. Sample size, complete statistical analysis, and $p$ values are provided in tables in supplementary datasets.

integrins in NIH-3T3 fibroblasts (Fig. 1g). Specifically, the paxillin/β3 integrin ratio was reduced to 60% and 45% in β3[VE] and β3[VE/YA] integrin-presenting FAs, respectively (Fig. 1g), confirming the previously reported diminished recruitment of paxillin to β3[VE/YA] integrin[44]. To understand why overexpressed mCherry_paxillin obliterated the differential recruitment of endogenous paxillin, we tested possible paxillin recruitment to endogenous β1 integrins[44], potentially "contaminating" β3 integrin-containing FAs. Substrate-bound β1 integrins were assessed by immunostaining with the 9EG7 mAb (Supplementary Fig. 2a–d). However, the relative amount of activated β1 integrin was similar in β3[WT] and β3[VE/YA] integrin-positive FAs (Supplementary Fig. 2a, b). Although the formation of some central β1 integrin-positive fibrillar adhesions was observed, paxillin preferentially localized to peripheral β3 integrin-containing FAs (Supplementary Fig. 2d). This result revealed specific β3 integrin recruitment to FAs when cells are cultured in complete medium, which contains vitronectin as the main spreading factor[53], and which was in striking contrast to β1 integrin-expressing C2C12 myoblasts plated on fibronectin-coated surfaces and cultured in 1% FBS-containing medium. In fact, in the latter cells, paxillin was recruited to β1 integrin-containing FAs and central fibrillar adhesions (Supplementary Fig. 2e). Thus, these data suggested that NIH-3T3 fibroblasts form a suitable experimental system in which signaling-defective β3 integrin mutants could be evaluated functionally and this setup provided a unique opportunity to elucidate the mechanisms of paxillin recruitment to αvβ3 integrin-containing adhesions, with negligible influence of endogenous β1 integrin.

Cells expressing GFP-tagged β3 integrins displayed adhesions whose morphology was influenced by the mutations introduced on the cytoplasmic tail of the β3 receptor, rather than by the cellular expression levels of paxillin (Fig. 1h–m and Supplementary Fig. 3a–c). In general, β3[VE/YA] integrin-expressing cells presented bigger (Fig. 1h, k and Supplementary Fig. 3a) and more elongated FAs (Fig. 1i, l and Supplementary Fig. 3b). Prominent GFP-positive central adhesions were induced by the high-affinity talin-binding motif (β3[VE]), likely accounting for the increased number of adhesions detected per field of acquisition (Fig. 1j, m and Supplementary Fig. 3c). In addition, live cell imaging of double-transfected NIH-3T3 cells unveiled a striking instability and accelerated retrograde sliding of β3[VE/YA] integrin-containing FAs, when compared to those presenting β3[WT] or β3[VE] integrins (Figs. 1n, 2a, and Supplementary Movies 1–3).

**Paxillin anchoring to the signaling-defective β3[VE/YA] integrin rescues the rapid sliding adhesion phenotype.** Aiming for a better understanding of the molecular mechanism underlying the rapid adhesions sliding phenotype of the signaling-defective β3[VE/YA] integrin (Fig. 2a), the F-actin network, and its retrograde movement were investigated. To this purpose, the flow of GFP-tagged LifeAct within β3 integrin-positive FAs was quantified using a frame-by-frame pixel-flux-based algorithm (Fig. 2b and Supplementary Fig. 3d). This revealed a correspondence between the motion of β3 integrins and actin within FAs (Fig. 2a, b), especially for the VE-containing–integrins, which talin-mediated connection with the actin network is likely to be stronger (actin/integrin flow ratio: β3[WT] 1.4, β3[VE] 1.1, and β3[VE/YA] 1.2). Based on these results it is conceivable that the tyrosine mutation has affected the ability of the receptor to firmly bind to its extracellular ligand, thus leading to its retrograde sliding together with the mechanically connected F-actin. Indeed, the sliding phenotype was rescued by introducing the ectodomain-activating N[305]T mutation into the β3[VE/YA] integrin (β3[N305T+VE/YA]) (Fig. 2a), suggesting that the VE/YA mutation undermined the inside-out activation of the integrin receptor. On the other hand, the rapid sliding of the β3[VE/YA] integrin could also be prevented by the co-expression of the wild-type β3 integrin (Fig. 2a, β3[VE/YA]_GFP + β3[WT]_mCh), proposing a threshold of wild-type or endogenous β3 integrin to rescue the sliding phenotype. Likewise, the slow flow of the β3[WT] integrin was not affected by the co-expression of the β3[VE/YA] integrin mutant within FAs (Fig. 2a, β3[WT]_GFP + β3[VE/YA]_mCh), suggesting that the high-affinity talin-binding site in the β3[VE/YA] integrin was not able to induce a dominant-negative effect by sequestering endogenous talin away from the β3[WT] integrin.

Although mCherry_paxillin properly localized at FAs, the loss of endogenous paxillin from β3 integrin mutants-containing FAs indicated a possible defect of either paxillin recruitment or binding to these receptors (Fig. 1g), which could be apparently rescued by its overexpression (Fig. 1b, d). Considering also the previously reported diminished recruitment of paxillin to β3[VE/YA] integrin[44], the enhanced apparent sliding of FAs presenting this receptor was intriguing. In fact, it contradicted current models proposing that an increased paxillin recruitment is associated with enhanced adhesion turnover[8,54]. We therefore assessed paxillin recruitment to β3[VE/YA]-adhesions by means of an alternative technique and determine whether it fulfills a critical role in activating the clutch

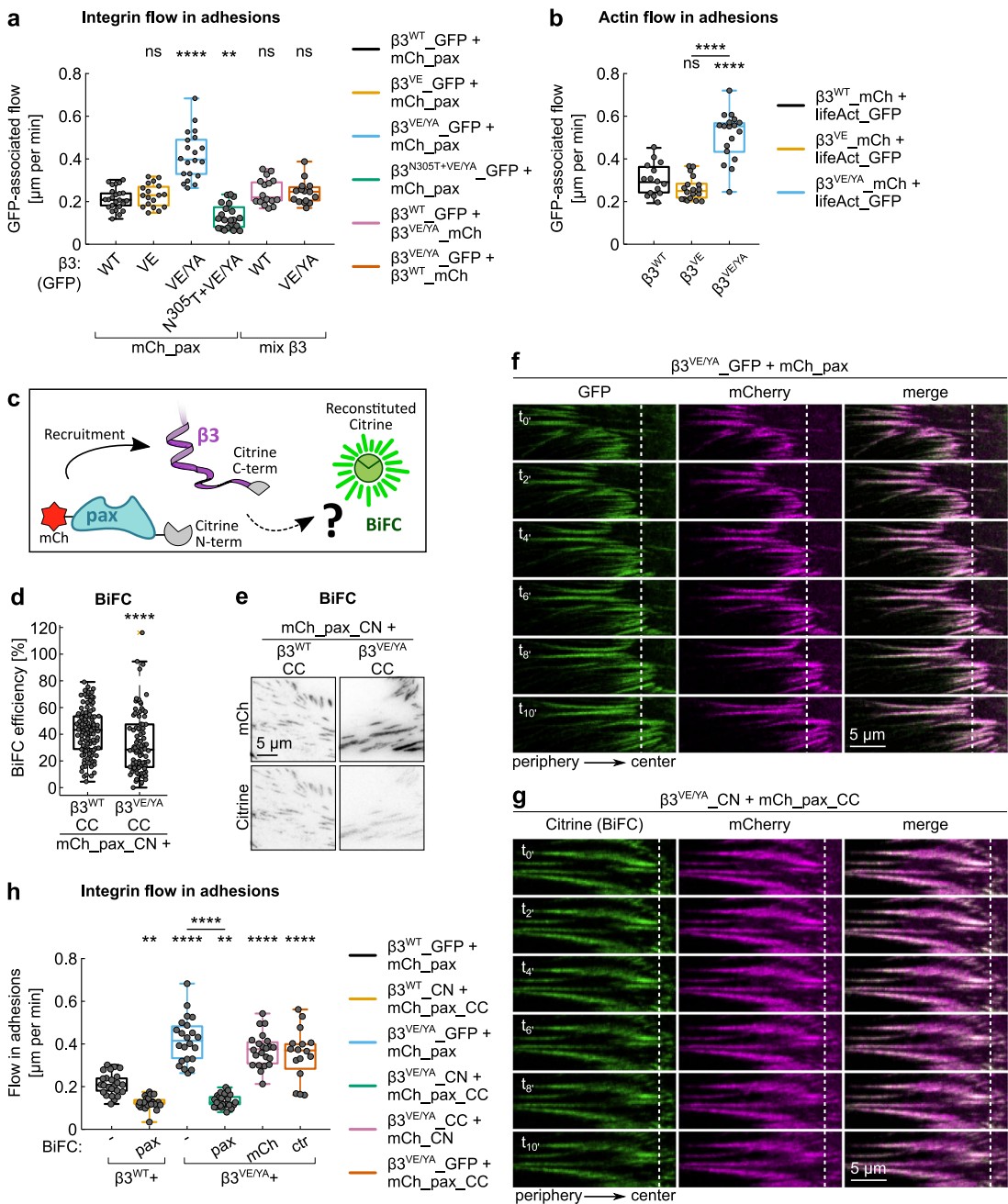

**Fig. 2 Dynamics of wild-type and mutant β3_GFP integrin-positive adhesions. a, h** Quantification of the β3 integrins flow in adhesions, expressed as mean displacement over time (μm/min) per replicate. Statistical analysis is provided in Supplementary Dataset 2. **b** Quantification of the actin flow in adhesions, expressed as mean displacement over time (μm/min) per replicate. Statistical analysis is provided in Supplementary Dataset 2. **c** Schematic representation of the BiFC assay used to evaluate paxillin recruitment to β3 integrins. **d, e** BiFC upon co-expression of mCherry_paxillin_CN and β3^WT_CC or β3^VE/YA_CC in NIH-3T3 fibroblasts. **d** Quantification of the BiFC efficiency. Statistical analysis is provided in Supplementary Dataset 3. **e** Representative TIRF images. Brightness and contrast were adjusted equally among different conditions. **f, g** Kymographs representative of the displacement of β3^VE/YA integrin-containing adhesions, over a time period of 10 min, in the absence (**f**) and in the presence (**g**) of mechanically (BiFC) coupled paxillin. ns (not significant) $p > 0.05$; *$p \leq 0.05$; **$p < 0.01$; ***$p < 0.001$; ****$p < 0.0001$. Sample size, complete statistical analysis, and $p$ values are provided in tables in supplementary datasets.

between the substrate-bound integrin and the retrogradely moving F-actin network[4].

Bimolecular Fluorescence Complementation (BiFC) was used as a tool to measure paxillin recruitment to β3 integrin-positive adhesions. Accordingly, mCherry_paxillin and β3^WT/β3^VE/YA integrins were C-terminally tagged with complementary citrine fragments (CN, citrine N-terminal fragment and CC, citrine C-terminal fragment) and co-expressed, as shown in Fig. 2c.

Transfected NIH-3T3 cells, recognized in virtue of their mCherry emission, were imaged via total internal reflection fluorescence (TIRF) microscopy, in order to detect the BiFC-derived citrine fluorescence. Although both combinations of proteins generated BiFC (Fig. 2d, e), the VE/YA mutation significantly reduced the efficiency. Swapping the citrine fragments among the proteins under study led to comparable results (Supplementary Fig. 4a), suggesting a diminished recruitment of paxillin to β3^VE/YA

integrin[44]. An alternative explanation of the reduced BiFC efficiency is that a proper paxillin recruitment was followed by an impaired and very transient paxillin binding to $\beta3^{VE/YA}$-containing adhesions, which reduced the probability of complementation of the citrine fragments[55–57]. Nevertheless, the generation of BiFC proposed that paxillin could be at least transiently recruited to $\beta3^{VE/YA}$ integrin-FAs, potentially relying on interactions with other FA-proteins such as kindlin or talin[16,37–39,44]. Strikingly, the establishment of a BiFC-based mechanical coupling of paxillin to $\beta3^{VE/YA}$ integrin rescued the FA-sliding phenotype (Fig. 2f, g and Supplementary Movie 4), confirmed by the quantification of the flow of the integrin-associated fluorescence (Fig. 2h). Indeed, the VE/YA mutation almost doubled the apparent displacement of integrins per minute ($\beta3^{WT}$ 0.21 μm/min, $\beta3^{VE/YA}$ 0.42 μm/min), but, after the BiFC-based trapping of paxillin, the stability was recovered ($\beta3^{VE/YA}$ + pax-BiFC 0.13 μm/min) and even improved compared to $\beta3^{WT}$ integrin-containing adhesions. Interestingly, in the presence of the mechanical integrin-paxillin coupling, the displacements of $\beta3^{WT}$ and $\beta3^{VE/YA}$ containing FAs were indistinguishable (Fig. 2h). When the BiFC probe of the $\beta3^{VE/YA}$ integrin was combined with the complementary citrine fragment presented by only the mCherry fluorophore ($\beta3^{VE/YA}$_CC + mCh_CN), adhesion sliding was not prevented, showing the paxillin-dependence for the rescue of this phenomenon. In addition, we excluded that the rescuing of the defective stability could simply rely on the expression of the double-tagged mCherry_paxillin_CC, as the flow of $\beta3^{VE/YA}$_GFP integrin was not modified in the absence of BiFC (Fig. 2h, $\beta3^{VE/YA}$_GFP + mCh_pax_CC). These results propose that the fast retrograde-sliding adhesion phenotype induced by the $\beta3^{VE/YA}$ integrin mutant was linked to a compromised paxillin binding and retention within FAs, possibly impairing the maturation of highly dynamic integrin/talin clusters into mature FAs[46].

**Photoactivation of paxillin reveals its rapid dissociation from $\beta3^{VE/YA}$ integrin-containing adhesions**. Following the hypothesis that impaired paxillin binding caused the instability of $\beta3^{VE/YA}$-positive FAs, paxillin dynamics at adhesion sites was investigated. Its dissociation rate was quantified via photoactivation experiments and used as a readout of its binding capacity to $\beta3$ integrin-positive FAs. Given the plausible influence of the VE and VE/YA mutations on the dynamics of talin, and therefore of $\beta3$ integrin itself[46], we first studied their influence on talin and integrin turnovers. To do so, $\beta3$ integrins were C-terminally tagged with the photoactivatable GFP (PA-GFP), co-expressed with mCherry_paxillin, and photoactivated in a single mCherry-positive peripheral FA. The fluorescence loss after photoactivation (FLAP) demonstrated that the high-affinity talin-binding motif (VE) enhanced the stability of the $\beta3$ integrin, prolonging its half-life ($t_{1/2}$) in adhesions by 1.4-fold (median $t_{1/2}$ of $\beta3^{WT}$ 104 s vs 147.1 s of $\beta3^{VE}$ integrin). However, for the $\beta3^{VE/YA}$ integrin mutant, the dissociation was accelerated, and the median of the estimated half-lives was 75.3 s, meaning 0.75-fold that of the wild-type integrin (Fig. 3a–c and Supplementary Fig. 5d).

To measure the talin off-rate, a similar photoactivation strategy was applied, with the $\beta3$ integrin carrying the mCherry tag. Coherent with the slower turnover of the $\beta3^{VE}$ integrin, the half-life of PA-GFP_talin was prolonged in $\beta3^{VE}$ compared to $\beta3^{WT}$ integrin-positive adhesions (82.6 and 46.7 s, respectively). Talin dissociation from $\beta3^{VE/YA}$ integrin adhesions (median $t_{1/2} = 43.2$ s) was instead almost identical to that observed with the wild-type receptor (median $t_{1/2} = 46.7$ s) (Fig. 3d–f and Supplementary Fig. 5e). Nevertheless, the ratio between half-lives of talin and integrins revealed a stabilization of talin on the

VE-expressing integrins: the half-life of talin in respect to that of $\beta3^{WT}$ integrin was 0.4, while increased to 0.6 in FAs containing $\beta3^{VE}$ or $\beta3^{VE/YA}$ integrins.

Considering the intricate network of paxillin interactions, we assured that the tagging of paxillin did not affect its binding capacity to $\beta3$ integrin-containing FA-complexes. The equivalent FLAP of recombinant paxillins presenting the PA-GFP either at the N- or C-terminus indicated that the fusion of a tag at the extremities did not alter its binding to $\beta3$ integrin-FAs (Supplementary Fig. 5a–c). Surprisingly, the VE motif, which increased the affinity of talin for the $\beta3$ integrin tail by about 20-fold[44] and reduced the dissociation rate of these two proteins from $\beta3$ integrin-presenting FAs, caused a faster dissociation of paxillin and a shortening of its half-life by ~40% (paxillin $t_{1/2}$ in $\beta3^{WT}$ adhesions 72 s, while in $\beta3^{VE}$ adhesions 42.3 s) (Fig. 3g–i and Supplementary Fig. 5f). Adding the $Y^{747}A$ substitution into the $\beta3^{VE}$ integrin sequence further destabilized paxillin binding to FAs and shortened its half-life by 70% (19.7 s) (Fig. 3g–i and Supplementary Fig. 5f). Therefore, the impact of the VE/YA mutation was much greater on the dynamics of paxillin than on those of talin (almost unaltered between $\beta3^{VE/YA}$ and $\beta3^{WT}$ adhesions) and $\beta3$ integrin ($t_{1/2}$ of $\beta3^{VE/YA}$ reduced by 25% compared to that of $\beta3^{WT}$). Importantly, these dynamic measures of paxillin dissociation would propose a highly compromised paxillin binding to $\beta3^{VE/YA}$ adhesions, which might be functionally linked to their rapid-sliding phenotype.

**Paxillin dissociation kinetics reflects the mechanical stability of isolated focal adhesions**. Since the rapid inward-sliding of $\beta3^{VE/YA}$ integrin-presenting FAs was potentially linked to the fast dissociation of paxillin from them, it appeared relevant to characterize the mechanical stability of the entire $\beta3$ integrin-positive FA-complex with an alternative approach. We therefore applied the protocol developed by Kuo et al. to isolate apparently intact FAs[58] and we revealed their mechanical integrity in respect to the expressed GFP-tagged $\beta3$ integrin construct. Consistent with the rapid displacement of $\beta3^{VE/YA}$ integrin-containing adhesions, these apparently mechanical fragile structures, did not survive the FA-isolation procedure (no GFP signal could be detected on coverslips). On the contrary, the shearing of the PM, the washing away of the cytosol with a jet of PBS[58], and subsequent fixation, allowed detecting isolated FAs containing $\beta3^{WT}$ or $\beta3^{VE}$ integrins in virtue of their fluorescence (Fig. 4a, b). Importantly, these isolated adhesive plaques were mCherry_paxillin-positive. Reasonably, the dissociation kinetics of the photoactivation studies reflected the strength of paxillin binding to FAs, which determined its ability to remain associated during the FA-isolation process. Indeed, while in intact cells, overexpressed paxillin was recruited to $\beta3^{WT}$ and $\beta3^{VE}$ adhesions equally well (Fig. 1d, f), in isolated FAs, the expression of the $\beta3^{VE}$ integrin variant caused a drop of paxillin/integrin fluorescence ratio of about 50% (Fig. 4a, b), which was consistent with the accelerated FLAP of paxillin from this type of adhesions (Fig. 3g, h). To assure that the mCherry tag and the exogenous paxillin expression did not affect its retention in FAs, cells were transfected only with integrin proteins, and the presence of endogenous paxillin in isolated adhesions was verified by immunostaining. As for the mCherry-labeled paxillin, the immunofluorescence signal decreased substantially (to 55%) in adhesions presenting the $\beta3^{VE}$ compared to the wild-type integrin (Fig. 4c, d). We also evaluated the binding of vinculin and demonstrated its reduction in $\beta3^{VE}$ integrin-containing, compared to $\beta3^{WT}$ integrin-containing, adhesions (Fig. 4e, f). This was unexpected considering the proposed talin-dependent recruitment and binding of vinculin to adhesions and the expression of an integrin with a high-

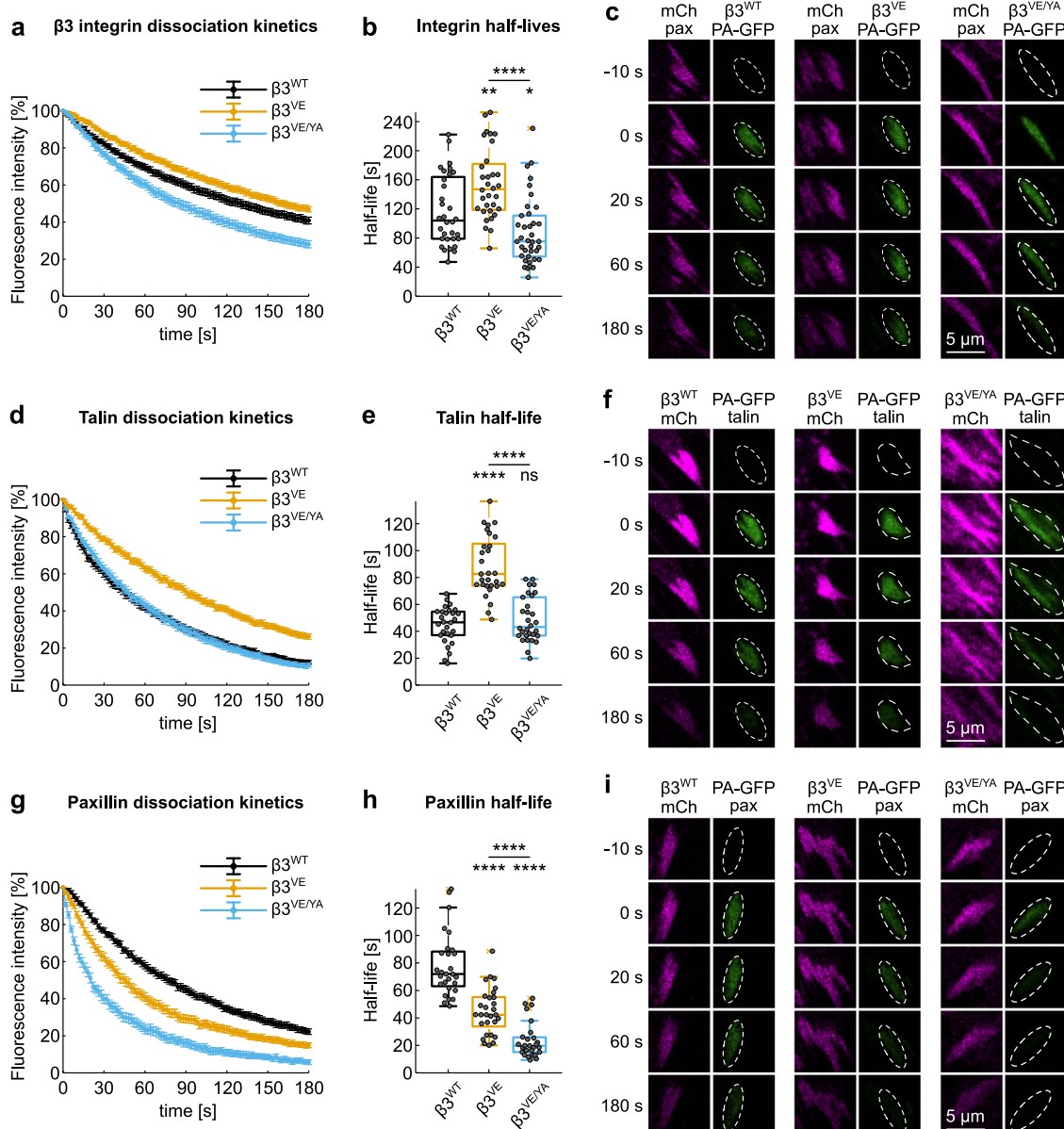

**Fig. 3 Analysis of the dissociation kinetics of the β3 integrin/talin/paxillin complex. a, d, g** Experimental dissociation kinetics of **a** β3_PA-GFP integrins from mCherry_paxillin-positive FAs and of **d** PA-GFP_talin and **g** PA-GFP_paxillin from β3_mCherry-positive FAs. **b, e, h** Box plot of the half-lives of **b** β3_PA-GFP integrins in mCherry_paxillin-positive FAs and of **e** PA-GFP_talin and **h** PA-GFP_paxillin in β3_mCherry-positive FAs. Statistical analysis is provided in Supplementary Dataset 4. **c, f, i** Representative images of photoactivation time series of experiments in **a, b, d, e**, and **f, g**. ns (not significant) $p > 0.05$; *$p \leq 0.05$; **$p < 0.01$; ***$p < 0.001$; ****$p < 0.0001$. Sample size, complete statistical analysis, and $p$ values are provided in tables in supplementary datasets.

affinity talin-binding site[2,4,40,44]. However, it was consistent with a critical role of paxillin in orchestrating the nanoscale localization of vinculin within FAs[10]. Accordingly, it has been demonstrated that paxillin and vinculin have comparable recovery curves after photobleaching[59], proposing that the VE mutation loosened vinculin binding to FAs, in a paxillin-dependent manner. These results suggested that, despite a high-affinity talin/integrin interaction, a proper paxillin-mediated FA assembly and maturation is required to firmly anchor cells to the substratum and to create adhesion complexes with sufficient stability to survive the isolation protocol. In line with these hypotheses, the N305T mutation within the β3VE/YA integrin (Fig. 4g), as well as the BiFC-mediated paxillin association to β3VE/YA integrin (Fig. 4h), allowed the proper isolation of β3 integrin-positive FAs.

**The N-terminal, LD motif-containing sequence, enhances paxillin stability in FAs.** By applying photoactivation and FA-extraction protocols to two paxillin fragments corresponding to the N-terminal LD motif-containing sequence and to the C-terminal LIM domains, respectively (Fig. 5a), we confirmed the previously proposed FA-targeting activity of the second[21]. Although the N-terminal paxillin fragment could be photoactivated and detected in the area corresponding to FAs (Supplementary Fig. 6a), it showed no specific retention and dissociated similarly to the cytoplasmically located PA-GFP control (median half-life of 1.8 and 2 s, respectively) (Fig. 5b, c and Supplementary Fig. 5g). Accordingly, this paxillin fragment was not detected in isolated adhesions above control levels (Fig. 5d, e). In contrast, the LIM domain-only construct (paxillin C-terminus) was efficiently photoactivated

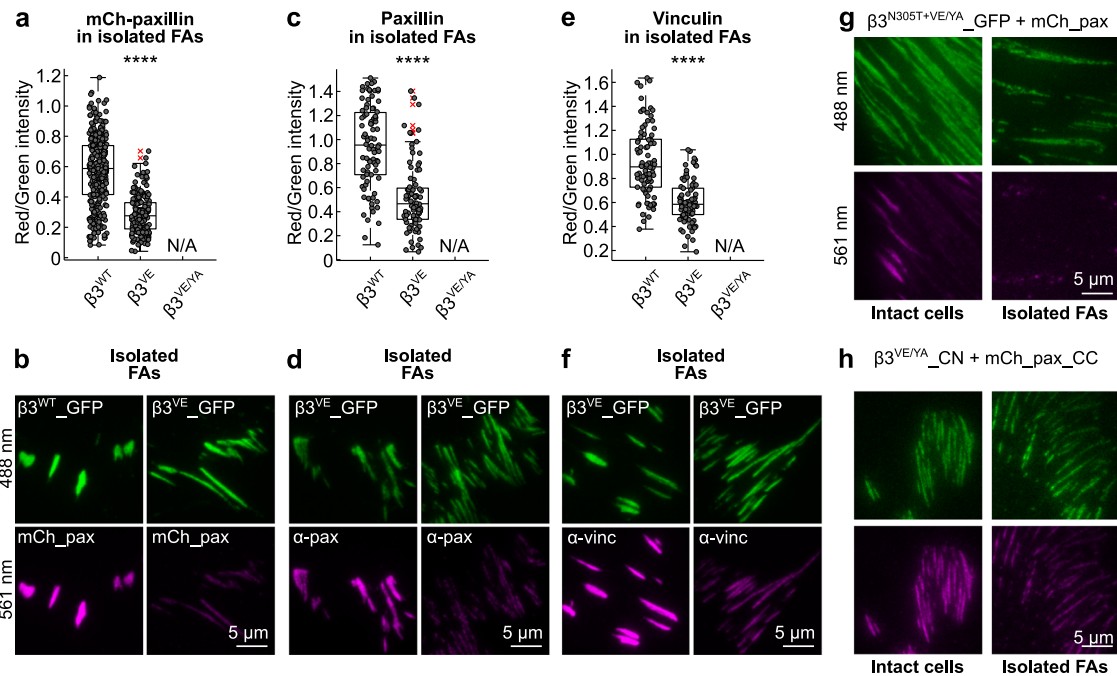

**Fig. 4 Analysis of paxillin and vinculin binding upon isolation of FA-complexes. a, c, e** Quantification of cytoplasmic proteins retained in β3$^{WT}$_GFP- or β3$^{VE}$_GFP-positive FAs, isolated from NIH-3T3 fibroblasts. **a** mCherry_paxillin, **c** endogenous paxillin, and **e** endogenous vinculin. Statistical analysis is provided in Supplementary Dataset 1. **b, d, f** Representative TIRF images of FAs isolated from NIH-3T3 cells of experiments quantified in **a, c,** and **e**. **g, h** Representative TIRF images of intact cells and FAs isolated from NIH-3T3 cells double transfected with **g** β3$^{N305T+VE/YA}$_GFP and mCherry_paxillin and **h** β3$^{VE/YA}$_CN and mCherry_paxillin_CC. ns (not significant) $p > 0.05$; *$p \leq 0.05$; **$p < 0.01$; ***$p < 0.001$; ****$p < 0.0001$. Sample size, complete statistical analysis, and $p$ values are provided in tables in supplementary datasets.

(Supplementary Fig. 6a), presented a half-life in FAs reduced to 50% compared to that of full-length paxillin (Fig. 5b, c and Supplementary Fig. 5g) and was significantly present in isolated β3$^{WT}$_GFP-positive adhesions (LIMs/integrin fluorescence 59% in respect to paxillin/integrin) (Fig. 5d, e). These data, therefore, propose a LIM domain-dependent localization of paxillin at β3 integrin-positive FAs, supported by an auxiliary function of the paxillin N-terminal segment.

To circumscribe this stabilization activity to a specific paxillin sequence, the dynamic behavior of δ paxillin, a naturally occurring paxillin isoform missing the first 133 amino acids (Fig. 5f), was investigated. Compared to LIMs-only, δ paxillin showed an improved FA retention (median $t_{1/2}$ 35.4 and 47.4 s, respectively), but still a dissociation rate significantly faster than that of the canonical paxillin (72 s) (Fig. 5g, h, Supplementary Figs. 5h and 6b). Interestingly, δ paxillin lacks tyrosines 31 and 118, which phosphorylation enhances FAK and vinculin binding[10,14,60]. When their phosphorylation was prevented by means of site-directed mutagenesis (paxillin Y$^{31/118}$F, Fig. 5f), the resulting dissociation kinetics overlaid with that of δ paxillin. These results are consistent with the hypothesis that paxillin stabilization in adhesion occurs in response to phosphorylation of the Y$^{31}$ and/or Y$^{118}$, potentially leading to an exposure of the LD motifs that enables an interaction with other components of FAs, such as talin, FAK, and vinculin[10,16,21,61,62].

Considering this stabilizer function of the paxillin N-terminus, we tested if the paxillin-dependent arrest of β3$^{VE/YA}$ integrin-positive FAs relied on its activity. Although the BiFC-based mechanical coupling of δ paxillin or paxillin Y$^{31/118}$F to the mutant integrin was as efficient as the wild-type protein, the LIM domains alone were, surprisingly, sufficient to arrest the inward flow of β3$^{VE/YA}$ integrin in adhesion to wild-type levels (β3$^{VE/YA}$ + LIMs BiFC 0.20 μm/min, β3$^{WT}$ without BiFC 0.21 μm/min) (Fig. 5i). This further corroborated a critical function of the LIM domains in mechanically

stabilizing integrin/talin complexes, thus suggesting an essential structural role for paxillin in β3 integrin adhesions. On the other hand, the deletion of the LIM domains and the conjugation of the remaining paxillin N-terminus to the β3$^{VE/YA}$ integrin via BiFC (β3$^{VE/YA}$_CN + mCherry_paxN-term_CC) was also sufficient to avoid the rapid inward sliding of this integrin mutant, raising the possibility of alternative mechanisms accounting for the stabilization of the β3$^{VE/YA}$ integrin, for example involving paxillin binding to small GTPases which control actin polymerization[17,18,46,63].

**Differential contribution of LIM domains to paxillin docking to FAs.** To precisely assess the contribution of each LIM domain to paxillin docking to FAs, we designed paxillin mutants missing individual LIM domains (Fig. 6a, b and Supplementary Fig. 7a), verified their proper expression (Fig. 6c and Supplementary Fig. 1c), and analyzed them by photoactivation and FA-isolation experiments. The first approach unveiled a wide range of paxillin dissociation dynamics (Fig. 6d, e, Supplementary Figs. 5i and 6c), strongly correlating with the quantitative analysis of their mechanical retention in isolated FAs (Fig. 6f, g). Specifically, the medians of the dissociation half-lives measured for paxillin-ΔLIM1, -ΔLIM2, -ΔLIM3, and -ΔLIM4 were, respectively, 15%, 7%, 9%, and 42% of that of paxillin wt (Fig. 6e). Similarly, the residual normalized paxillin-derived fluorescence in isolated adhesions were 14%, 7%, 10%, and 25% compared to that of mCherry_paxillin wt (Fig. 6f). In line with previous qualitative observations[21,36], the deletion of LIM4 was significantly the least severe, while surprisingly the LIM1 and LIM2 deletions were comparable to that of LIM3, which so far represented the most critical FA-interaction domain. These results suggested the existence of multiple FA-interaction sites, spread along the four LIM domains. Except for LIM4, which removal would not affect the spacing between LIM1–LIM2–LIM3 domains, the outcome of all other LIM-deletions could also reflect the mis-localization of the

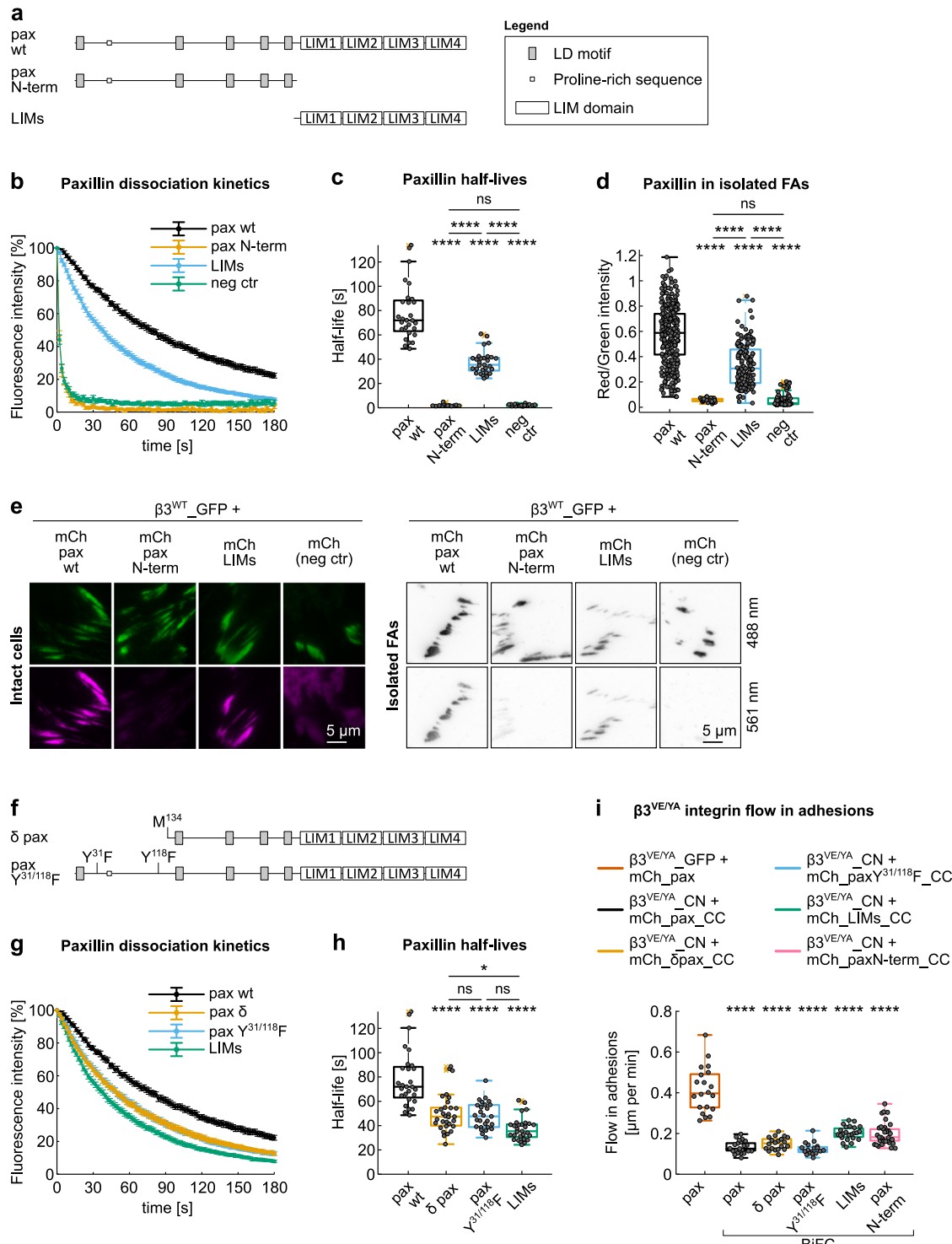

**Fig. 5 Paxillin binding to adhesions is driven by the LIM domains but stabilized by the N-terminus. a**, **f** Schematic representation of paxillin **a** truncation mutants and **f** N-terminal mutants. All proteins were N-terminally tagged with PA-GFP or mCherry. **b**, **g** Experimental dissociation kinetics of PA-GFP_paxillin wt and **b** truncation mutants or **g** N-terminal mutants, from β3_mCherry-positive FAs. **c**, **h** Box plot of the half-life of PA-GFP_paxillin wt and **c** truncation mutants or **h** N-terminal mutants, in β3_mCherry-positive FAs. Statistical analysis is provided in Supplementary Dataset 4. **d** Quantification of mCherry_paxillin wt and truncation mutants retained in FAs, isolated from NIH-3T3 cells co-expressing β3WT_GFP. Statistical analysis is provided in Supplementary Dataset 1. **e** Representative TIRF images of FAs in NIH-3T3 cells co-expressing β3WT_GFP and mCherry_paxillin proteins. Left panel: FAs in intact cells; right panel: isolated FAs. **i** Quantification of the β3VE/YA integrin flow in adhesions, expressed as mean displacement over time (μm/min) per replicate. Statistical analysis is provided in Supplementary Dataset 2. Neg ctr: PA-GFP in photoactivation experiments and mCherry in isolated FAs. ns (not significant) $p > 0.05$; *$p \leq 0.05$; **$p < 0.01$; ***$p < 0.001$; ****$p < 0.0001$. Sample size, complete statistical analysis, and $p$ values are provided in tables in supplementary datasets.

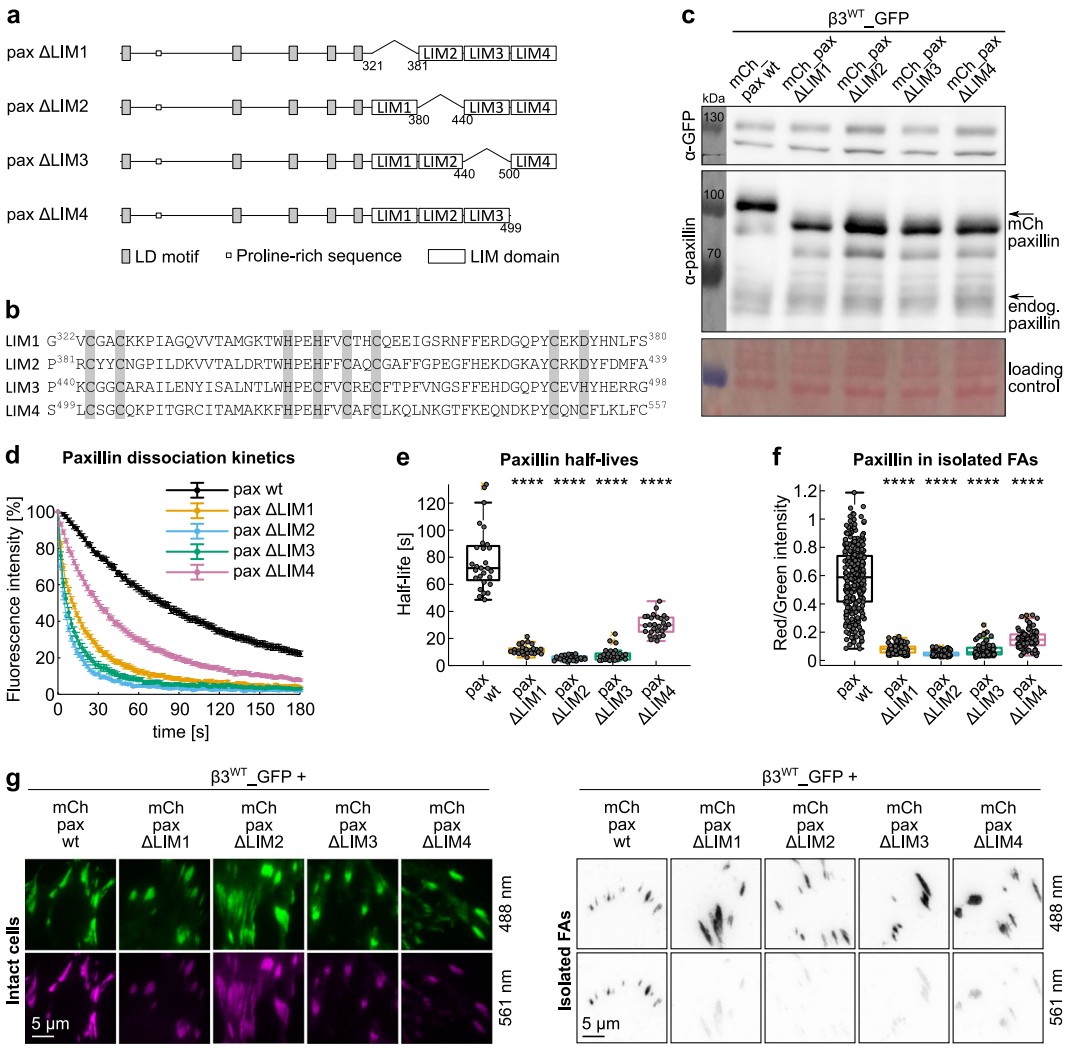

**Fig. 6 LIM domain deletion analysis by photoactivation and FA-isolation. a** Schematic representation of the paxillin LIM domain deletion mutants. All proteins were N-terminally tagged with PA-GFP or mCherry. **b** Alignment of the four paxillin LIM domains. Gray: zinc-binding residues. **c** Western blot of protein extracts from NIH-3T3 fibroblasts transiently co-expressing β3^WT_GFP and mCherry-tagged paxillin proteins, wt or deletion mutants. Loading control: Ponceau staining. **d** Experimental dissociation kinetics of PA-GFP_paxillin wt and deletion mutants from β3^WT_mCherry-positive FAs. **e** Box plot of the corresponding half-lives Statistical analysis is provided in Supplementary Dataset 4. **f** Quantification of mCherry_paxillin wt and mutants retained in FAs isolated from NIH-3T3 cells co-expressing β3^WT_GFP. Statistical analysis is provided in Supplementary Dataset 1. **g** Representative TIRF images of FAs in NIH-3T3 cells co-expressing β3^WT_GFP and mCherry_paxillin proteins. Left panel: FAs in intact cells; right panel: isolated FAs. ns (not significant) $p > 0.05$; *$p \leq 0.05$; **$p < 0.01$; ***$p < 0.001$; ****$p < 0.0001$. Sample size, complete statistical analysis, and $p$ values are provided in tables in supplementary datasets.

remaining LIM domains, preventing to identify the specific binding function of the missing LIM domain.

Given the possibility that each paxillin LIM domain constitutes a binding site for FA components, we assessed whether the position of N-terminal PA-GFP tag has potentially impaired these interactions. Accordingly, the fluorescent protein was further insulated from LIM1 by including the LD5 sequence or relocated at the C-terminus of the LIMs-only construct, and photoactivation experiments were carried out (Supplementary Fig. 5a–c). While no differences were detected in the dissociation rates of LIMs and LD5+LIMs N-terminally tagged recombinant proteins, indicating that the photoactivatable GFP unlikely compromised LIM1 accessibility, a slight but relevant acceleration of LIMs dynamics was caused by their tagging at the C-terminus, further confirming that the so far ignored LIM4 has indeed a role in the paxillin association to FAs.

**LIM domains are not interchangeable and LIM2 is the major determinant of paxillin docking to FAs.** Because of the confounding factor generated by the LIM domain deletion strategy, which abolished the function of a LIM domain but at the same time also destroyed their respective spacing (with the exception of LIM4 truncation), we developed the LIM domain replacement strategy (Fig. 7a, b and Supplementary Fig. 7b).

A specific LIM1 domain-associated function was revealed by replacing it by either LIM2 or LIM3 and assessing the dissociation kinetics of the resulting paxillin 2234 and paxillin 3234 proteins (Fig. 7a) from β3 integrin-positive adhesions (Fig. 7c, d, Supplementary Figs. 5j and 6d). While both mutants partially rescued the deletion of LIM1, none of them presented a level of binding to β3 integrin adhesions comparable to that of paxillin wt, indicating a missing LIM1-dependent interaction. Nevertheless, LIM2 was slightly more efficient than LIM3 in rescuing

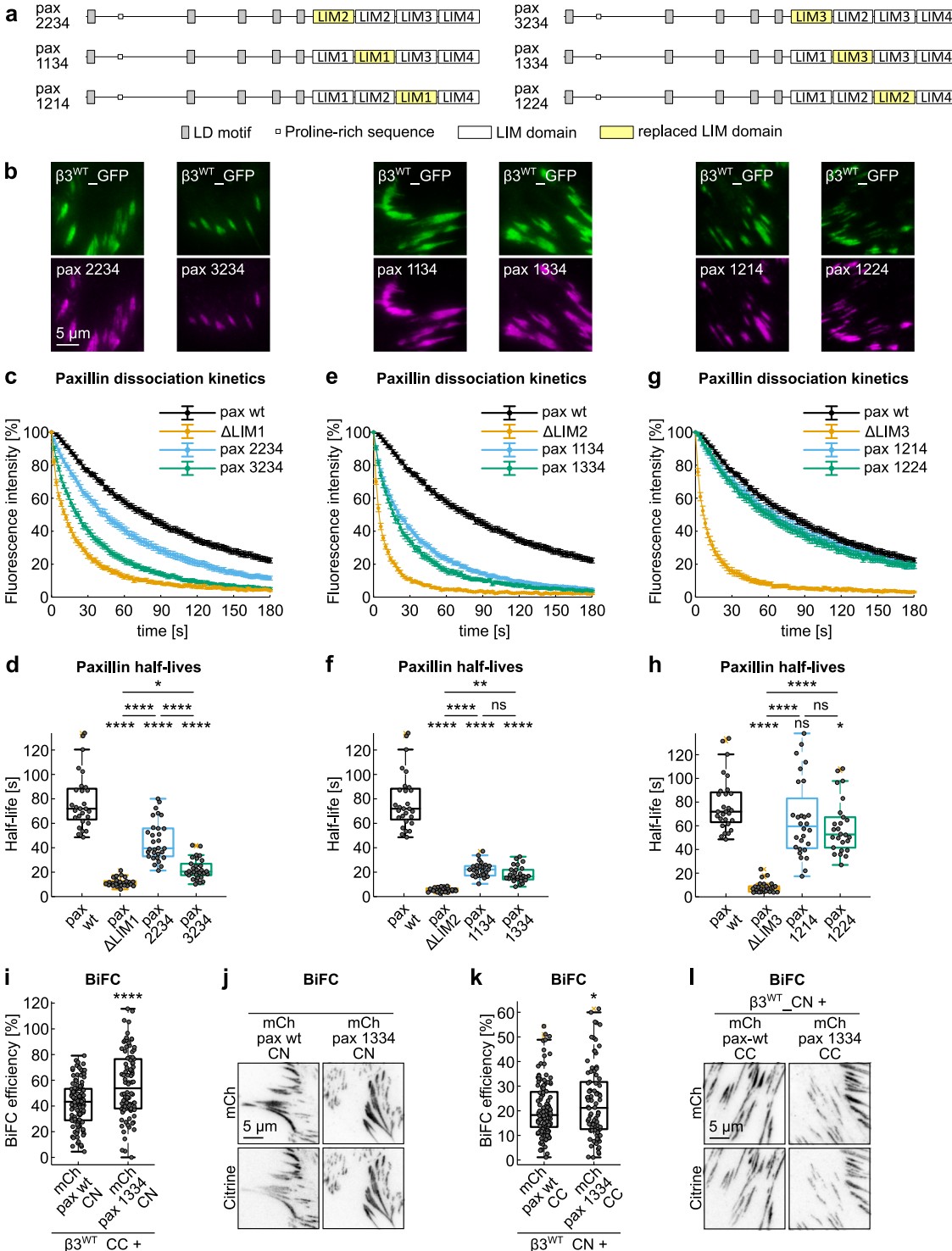

**Fig. 7 LIM domain-specific functions assessed by the LIM domain replacement strategy. a** Schematic representation of the paxillin mutants obtained by the LIM domain replacement strategy. From the top line: replacement of LIM1, replacement of LIM2, and replacement of LIM3. All proteins were N-terminally tagged with fluorophores. **b** Representative TIRF images of NIH-3T3 co-expressing β3^WT_GFP and mCherry-tagged paxillin variants with replaced LIM domains. **c, e, g** Experimental dissociation kinetics of PA-GFP_paxillin wt and **c** LIM1 mutants, **e** LIM2 mutants, and **g** LIM3 mutants, from β3_mCherry-positive FAs. **d, f, h** Box plot of the half-lives of PA-GFP_paxillin wt and **d** LIM1 mutants, **f** LIM2 mutants, and **h** LIM3 mutants in β3_mCherry-positive FAs. Statistical analysis is provided in Supplementary Dataset 4. **i, j** BiFC upon co-expression of mCherry_paxillin_CN or its corresponding paxillin 1334 mutant and β3^WT integrin C-terminally tagged with the citrine C-terminal fragment in NIH-3T3 cells. **i** Quantification of the BiFC efficiency. Statistical analysis is provided in Supplementary Dataset 3. **j** Representative TIRF images. Brightness and contrast were automatically adjusted. **k, l** BiFC upon co-expression of mCherry_paxillin_CC or its corresponding paxillin 1334 mutant and β3^WT integrin C-terminally tagged with the citrine N-terminal fragment in NIH-3T3 cells. **k** Quantification of the BiFC efficiency. Statistical analysis is provided in Supplementary Dataset 3. **l** Representative TIRF images. Brightness and contrast were automatically adjusted. ns (not significant) $p > 0.05$; *$p \leq 0.05$; **$p < 0.01$; ***$p < 0.001$; ****$p < 0.0001$. Sample size, complete statistical analysis, and $p$ values are provided in tables in supplementary datasets.

LIM1 deletion, proposing that despite the high degree of homology among their primary structure, LIM2 presents some residues that better mimicked the precise and unique signature of LIM1.

The half-life of paxillin lacking LIM2 (paxillin ΔLIM2, 5.1 s) was only slightly improved by duplicating LIM1 (paxillin 1134, 22.1 s) or LIM3 (paxillin 1334, 16.4 s) (Fig. 7e, f, Supplementary Figs. 5k and 6d). These data favor the hypothesis that the LIM2 domain does possess exclusive features and irreplaceable functions that none of the other structurally similar paxillin LIM domains could execute.

When based on the sole analysis of the LIM domain deletion mutants, we would have expected that LIM3 is carrying key determinants of paxillin binding to β3 integrin-containing FAs, confirming the previously proposed findings by Brown et al.[21]. However, the LIM3 replacement by either LIM1 or LIM2, caused no significant (paxillin 1214) and only a minor (paxillin 1224) acceleration of paxillin dissociation from adhesions, when compared to the wild-type protein (Fig. 7g, h, Supplementary Figs. 5l and 6d). This corresponds to an efficient rescuing of the LIM3 deletion and proposes that a LIM3 domain-specific function is unlikely to be sequence-dependent, but rather relying on its overall 3D-folding, faithfully reproduced by both LIM1 and LIM2. In fact, it raises the possibility that LIM3 has an important role in creating the correct distance between LIM1–LIM2 and LIM4 domains, assuring their correct spatial separation when binding to the β3 integrin-positive FA-complex.

Finally, relying on a BiFC approach, we demonstrated that the paxillin binding to β3 integrin adhesions, but not its recruitment was altered by the LIM domain replacement strategy. Emblematic of these two distinct phenomena is the analysis of paxillin 1334, which was among the least retained mutants (Fig. 7e, f), showing dissociation kinetics comparable to that of wild-type paxillin from β3$^{VE/YA}$ integrin-containing FAs (Fig. 3g, h), yet accomplishing a BiFC of β3$^{WT}$ integrin at least as efficient as paxillin wt (Fig. 7i, j). Consistent BiFC results were obtained upon exchange of the citrine fragments among paxillin and integrin (Fig. 7k, l), confirming that the replacement of the LIM2 by LIM3 domain resulted in a protein (paxillin 1334) unable to make stable interactions with β3 integrin-positive FAs, which was, however, well, but only transiently, recruited to these sites.

Overall, these data revealed that individual LIM domains have specific FA-interaction functions linked to their respective position in the LIM domain repeats. Moreover, the efficient BiFC between paxillin 1334 and the C-terminus of β3 integrin suggests that despite reduced retention in FAs and the absence of the pivotal LIM2 domain, the mechanisms of recruitment of the LIMs-array in respect to the C-terminal end of the integrin tail was not altered.

**Defective paxillin recruitment to β3 integrin adhesions is not influenced by endogenous paxillin**. Above, we have shown that endogenous β3 and β1 integrins are likely not influencing paxillin recruitment and binding to β3_GFP-positive FAs. Similarly, endogenous paxillin did not affect the relative recruitment of exogenous wild-type paxillin to β3 integrin-containing FAs (Fig. 1d, f). However, in the presence of a FA-targeting deficient paxillin, the relative amount of the transfected mutant protein as compared to endogenous wild-type paxillin might influence the FA-recruitment process. We therefore tested the dissociation rate of paxillin 1334 from β3 integrin-positive FAs in paxillin-deficient fibroblasts (Supplementary Fig. 8a–c). Importantly, dissociation kinetics were essentially overlapping with those obtained in cells expressing endogenous paxillin, proposing that this dynamic measuring method is not sensitive to endogenous wild-type

paxillin. This notion was further confirmed, when assessing the ratio between mCherry_paxillin variants and β3_GFP in FAs in NIH-3T3 and paxillin-null fibroblasts by TIRF microscopy (Supplementary Fig. 8d–g). In particular, a drop of the relative fluorescence intensity in FAs was observed for the highly dynamic paxillin ΔLIM2 and paxillin 1334 proteins, while the LIM domain-only construct was comparable to paxillin wt, independently of the presence of endogenous paxillin. However, when measuring the relative fluorescence intensity of mCherry_paxillin variants with overexpressed wild-type GFP_paxillin in GFP-positive FAs of β3 integrin-containing Swiss-3T3 cells, the differences between the wild-type and LIM domain-only constructs and the recruitment deficient paxillin mutants were enhanced (Supplementary Fig. 8h, i). This clearly demonstrated the lower capacity of paxillin ΔLIM2 and paxillin 1334 constructs to compete with the wild-type full-length paxillin for binding to endogenous β3 integrins in Swiss-3T3 cells (Supplementary Fig. 8i), showing that the intracellular concentration, but also a limiting number of low or high-affinity binding sites in FAs affect the apparent localization of paxillin in cells.

**Paxillin LIM4 is a potential membrane binding and palmitoylated domain**. By sequence comparisons of the paxillin LIM domains, Beckerle and co-workers grouped the homologous LIM1, LIM2, and LIM3 but classified LIM4 differently[22]. When analyzed for specific features, LIM4 stands out for its remarkably high positive charge (+5 compared to −1, −1, and −2 of LIM1, LIM2, and LIM3, respectively). LIM4 is also unique for the presence of two conserved cysteines that do not coordinate the binding of zinc atoms (Cys$^{512}$ and the C-terminal Cys$^{557}$) and remain therefore potentially accessible for post-translational modifications (Fig. 8a). Based on these peculiar and previously unnoticed features we investigated a possible membrane-binding function for the paxillin LIM4 domain, intended to orient and stabilize paxillin in β3 integrin-presenting FAs. In fact, the cluster of basic residues could establish nonspecific electrostatic interactions with acidic phospholipids of the PM while the above-mentioned cysteines can be S-acylated (i.e., palmitoylated), further strengthening this membrane association.

Considering the link between protein structure/function and selective pressure over evolution, an alignment of paxillin LIM4 domain across species was performed (Fig. 8a). It appeared that this domain is highly conserved, that the C-terminal cysteine Cys$^{557}$, compared to Cys$^{512}$, arose later during evolution, and that much simpler organisms, although containing several sequence modifications, presented a perfectly intact GRC$^{512}$ motif and an overall high net positive charge (Fig. 8a). As these well-preserved features may be pivotal for the functioning of the domain, we sought to determine their contribution to paxillin retention in β3 integrin adhesions. Site-directed mutagenesis, preventing the hypothesized palmitoylation at Cys$^{512/557}$ or abrogating LIM4 positive charge, did not affect paxillin expression levels (Fig. 8b and Supplementary Fig. 1d) or paxillin localization to adhesions (Fig. 8c). However, they reduced the half-life of paxillin in β3 integrin-positive FAs, as shown by photoactivation experiments (Fig. 8d). Ensuring lipidation of the paxillin C$^{512/557}$S double mutant, by adding to the paxillin C-terminus a CaaX-box (PA-GFP_paxillinC$^{512/557}$S-CAIL), rescued paxillin FA-dissociation kinetics to wild-type levels (Fig. 8d). Interestingly, the mutation of the arginine within the conserved GRC motif (R$^{511}$Q) was comparable to the C$^{512}$S mutation (Fig. 8d), suggesting that this Arg$^{511}$ residue (and Gly$^{510}$ as well) may function to allow the lipidation of the following cysteine. The neutralization of paxillin LIM4 positive charge, by means of the RKK$^{511/532/554}$QED triple mutation, reduced the retention of paxillin in β3 integrin-positive

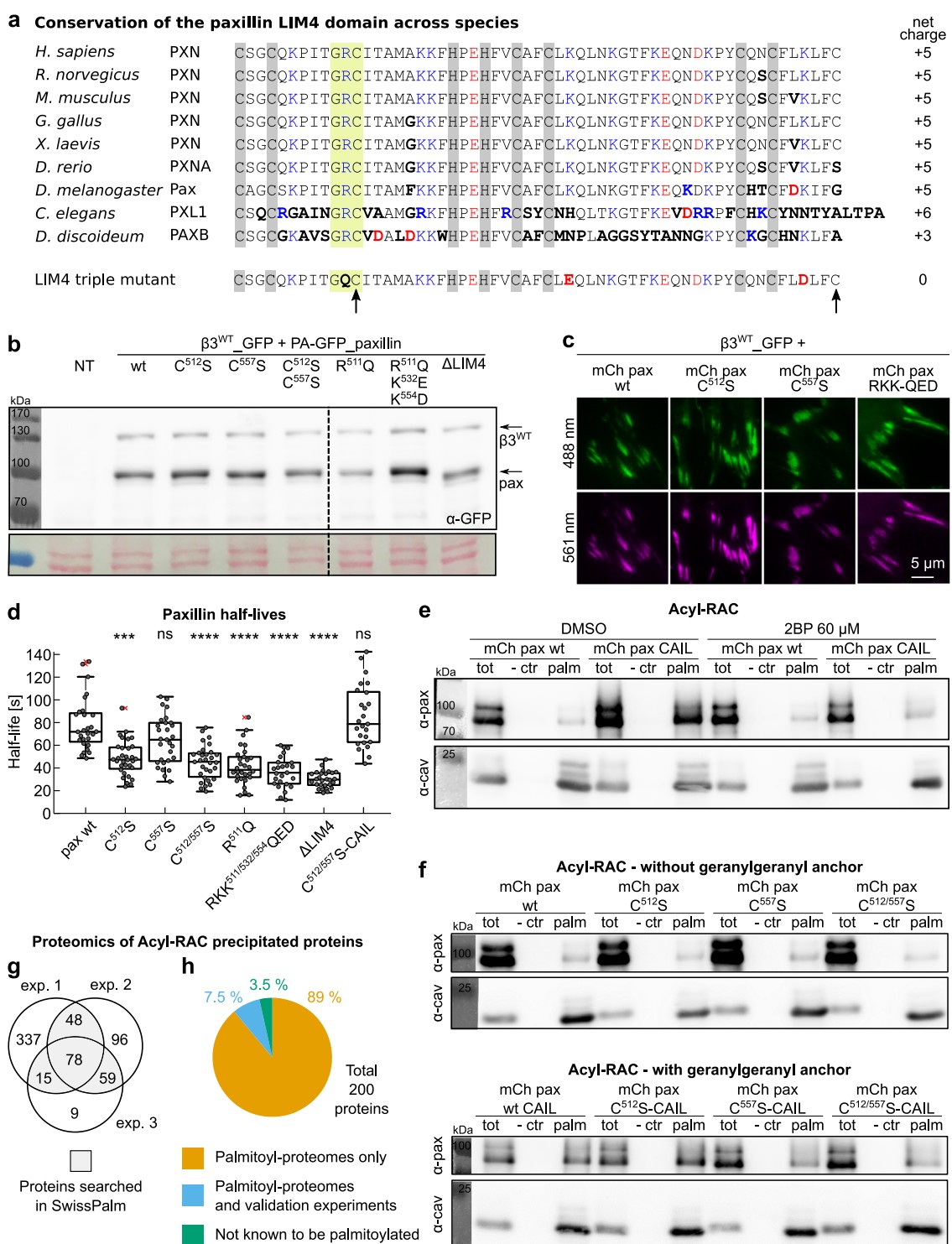

**a Conservation of the paxillin LIM4 domain across species**

| | | | net charge |
|---|---|---|---|
| *H. sapiens* | PXN | CSGCQKPITGRCITAMAKKFHPEHFVCAFCLKQLNKGTFKEQNDKPYCQNCFLKLFC | +5 |
| *R. norvegicus* | PXN | CSGCQKPITGRCITAMAKKFHPEHFVCAFCLKQLNKGTFKEQNDKPYCQSCFLKLFC | +5 |
| *M. musculus* | PXN | CSGCQKPITGRCITAMAKKFHPEHFVCAFCLKQLNKGTFKEQNDKPYCQSCFVKLFC | +5 |
| *G. gallus* | PXN | CSGCQKPITGRCITAMGKKFHPEHFVCAFCLKQLNKGTFKEQNDKPYCQNCFLKLFC | +5 |
| *X. laevis* | PXN | CSGCQKPITGRCITAMGKKFHPEHFVCAFCLKQLNKGTFKEQNDKPYCQNCFVKLFC | +5 |
| *D. rerio* | PXNA | CSGCQKPITGRCITAMGKKFHPEHFVCAFCLKQLNKGTFKEQNDKPYCQSCFVKLFS | +5 |
| *D. melanogaster* | Pax | CAGCSKPITGRCITAMFKKFHPEHFVCAFCLKQLNKGTFKEQKDKPYCHTCFDKIFG | +5 |
| *C. elegans* | PXL1 | CSQCRGAINGRCVAAMGRKFHPEHFRCSYCNHQLTKGTFKEVDRRPFCHKCYNNTYALTPA | +6 |
| *D. discoideum* | PAXB | CSGCGKAVSGRCVDALDKKWHPEHFVCAFCMNPLAGGSYTANNGKPYCKGCHNKLFA | +3 |
| | | | |
| LIM4 triple mutant | | CSGCQKPITGQCITAMAKKFHPEHFVCAFCLEQLNKGTFKEQNDKPYCQNCFLDLFC | 0 |

**b** β3^WT_GFP + PA-GFP_paxillin

**c** β3^WT_GFP +

**d Paxillin half-lives**

**e Acyl-RAC**

**f Acyl-RAC - without geranylgeranyl anchor**

Acyl-RAC - with geranylgeranyl anchor

**g Proteomics of Acyl-RAC precipitated proteins**

**h**
- Palmitoyl-proteomes only
- Palmitoyl-proteomes and validation experiments
- Not known to be palmitoylated

FAs to the same extent as the deletion of the entire LIM4 domain (Fig. 8d), demonstrating that its function relies on its positive charge, potentially supporting a PM interaction.

The Acyl-Resin Assisted Capture assay (Acyl-RAC), a valid tool to detect palmitoylated proteins among a total cell lysate, revealed a small but constant fraction of paxillin in the pool of palmitoylated proteins (Fig. 8e and Supplementary Fig. 1e). The amount of palmitoylated paxillin strongly increased when an irreversible geranylgeranyl lipid anchor was coupled to its carboxyl-terminus by means of the previously exploited CaaX-box (mCherry_paxilin-CAIL). Only in this context we

appreciated the effect of the treatment with 2-bromopalmitate (2BP) that, competing with endogenous palmitate, acts as a general inhibitor of palmitate incorporation. Caveolin-1, because of its essentially irreversible palmitoylation[64] is not sensitive to 2BP treatment and was therefore used as an internal positive control. As data proved that paxillin is palmitoylated, the next step was the attempt of identifying the palmitoylation site(s). Accordingly, the same technique was used to assess whether the Cys-to-Ser paxillin mutations, previously studied by means of the photoactivation technique, prevented the post-translational modification under investigation. Independently of the presence

**Fig. 8 Paxillin LIM4 domain carries membrane-binding features relevant for FA binding. a** Alignment of the amino acid sequence of paxillin LIM4 across species and triple human paxillin LIM4 mutant (bottom). The latter was obtained by replacing three basic amino acids with neutral or acidic residues found at the corresponding positions in other paxillin LIM domains (RKK[511/532/554]QED). Gray: zinc-binding residues; blue: positive residues; red: negative residues; bold: non-conserved residues in respect to the human paxillin LIM4 domain; arrows indicate the conserved free cysteines; yellow: conserved GRC motif. **b**, **c** Expression analysis of paxillin LIM4 mutants **b** via western blot and **c** TIRF microscopy. Loading control: Ponceau staining. **d** Box plot of the half-lives of PA-GFP_paxillin wt and LIM4 mutants in β3[WT]_mCherry-positive FAs. Statistical analysis is provided in Supplementary Dataset 4. **e**, **f** Acyl-Resin Assisted Capture assays, **e** from cell transiently transfected with mCherry_paxillin wt or mCherry_paxillin-CAIL, untreated (DMSO) or treated with 2-bromopalmitate and **f** from cell transiently transfected with mCherry_paxillin wt or Cys-mutants, (top) without and (bottom) with the CaaX-box. **g**, **h** Proteomics analyses of proteins recovered with the Acyl-RAC. **g** Venn Diagram of proteins identified in the three samples. **h** Result of the search in the SwissPalm database of proteins identified in at least two proteomic analyses. Complete protein list is provided in Supplementary Dataset 5. NT: not transfected; tot: total cell lysate; − ctr: negative control (tris-base); palm: fraction of palmitoylated proteins (hydroxylamine); 2BP: 2-bromopalmitate. ns (not significant) $p > 0.05$; *$p \leq 0.05$; **$p < 0.01$; ***$p < 0.001$; ****$p < 0.0001$. Sample size, complete statistical analysis, and $p$ values are provided in tables in supplementary datasets.

of an irreversible C-terminal lipid anchor, palmitoylation of any of the paxillin mutants was still possible (Fig. 8f and Supplementary Fig. 1f, g). In addition, as observed before, the geranylgeranylation obtained by the processing of the -CAIL sequence, enhanced paxillin palmitoylation ratio. These data indicated that at least one palmitoyl-cysteine was still present in each of the proteins exogenously expressed, preventing to conclude about the palmitoylation of the cysteines 512 and 557. To validate the Acyl-RAC results, proteins recovered in the palmitoylated fraction from three independent experiments were subjected to proteomic analysis and the following comparison with proteins in the SwissPalm database[65]. Among murine proteins recovered in at least two out of three experiments (Fig. 8g), 96.5% were annotated in the database as *S*-acylated proteins (Fig. 8h and Supplementary Dataset 5).

Overall, these data highlighted the relevance of LIM4 positive charges and demonstrated the palmitoylation of the paxillin protein, which is in line with a possible membrane-binding function of the LIM4 domain.

**BiFC shows a defined spatial orientation of paxillin LIM domains within β3 integrin-positive FAs.** A possible function of the proposed membrane-binding activity of LIM4, is to consolidate the specific orientation of the LIM domain-array within β3 integrin-containing FAs. Previous iPALM studies revealed a close membrane association of both paxillin termini[66]. However, based on various structures of two tandemly-organized LIM domains (e.g. PDB ID: 1RUT and PDB ID: 2DFY), it is conceivable that the paxillin LIM1–LIM2–LIM3–LIM4 array has an extended conformation. Because the BiFC efficiency is imposed by the relative position of the proteins under analysis, as well as by the spatial orientation of the complementary fluorophore fragments, we attempted to reveal the orientation of paxillin in respect to the PM by BiFC. Accordingly, as summarized in Fig. 9a, differentially tagged recombinant paxillins were tested for their ability to complement, in the context of β3 integrin-expressing fibroblasts (Swiss-3T3[44]), a membrane-anchored citrine fragment so localized via the PM-targeting peptide of Lck[67]. Despite the flexible nature of the paxillin amino-terminal half[68], only the paxillin C-terminus appeared close enough to the lipid bilayer for efficient BiFC with the PM-residing complementary probe (Fig. 9b, c). Importantly, the spatial orientation of paxillin was encoded by the LIM domains, which yielded comparable results in the absence of the paxillin N-terminus (Fig. 9b, c). The swapping of citrine fragments between the PM probe and paxillin/LIMs (Supplementary Fig. 4b) produced analogous outcomes, confirming the preferred complementation of the PM-localized CN fragment by C-terminally over N-terminally CC-tagged paxillin/LIMs proteins (Supplementary Fig. 4c). Interestingly, despite the general distribution of the PM-

anchored citrine fragments within the lipid bilayer (Supplementary Fig. 4e), the BiFC-driven paxillin-PM association identified in virtue of the citrine fluorescence, did not affect the specific FA localization of paxillin (Fig. 9c). On the other hand, in the few cells that showed an interaction between the N-terminus of LIMs and the PM probe, the BiFC signals were mainly localized in the PM outside of FAs (Supplementary Fig. 4f), suggesting that when trapped in an "inverted" orientation, LIM domains have a low affinity for FAs (Supplementary Fig. 4g).

To gather further insights about the mechanism of paxillin binding to the FA-complex, we mapped the relative position of the paxillin extremities compared to the β3 integrin tail, applying a similar BiFC strategy (Fig. 9a). Efficient complementation of CC-tagged β3[WT] integrin was obtained only when the matching citrine fragment was fused to the C-terminus of paxillin (Fig. 9d, e). Similarly, the BiFC efficiencies of the LIMs-only constructs were comparable to that of the corresponding full-length protein (Fig. 9d, e) and insensitive to the swapping of the complementary citrine fragments (Supplementary Fig. 4b), demonstrating the significantly higher interaction of the paxillin C-terminus with the integrin cytoplasmic tail (Supplementary Fig. 4d).

Although the sliding of β3[VE/YA] integrin adhesions was rescued by directly joining paxillin and the integrin receptor via BiFC (Fig. 2h), it is still controversial whether paxillin mediates FA-stabilization by direct β integrin binding[21,50,51]. We therefore asked whether an integrin-independent binding of paxillin could also rescue the sliding of the β3[VE/YA] integrin. Leveraging the disclosed knowledge about the structural organization of adhesion complexes, an alternative, LIM4-mediated paxillin-FA interaction was stabilized, in order to evaluate its potential to rescue the sliding. The irreversible anchoring of LIM4 to the PM was achieved by using the SpyTag/SpyCatcher technology[69] as presented in Fig. 9f. Upon co-expression of mCherry_paxillin_-SpyTag and PM_SpyCatcher a spontaneous reaction results in a stable isopeptide bond, which covalently conjugated the two recombinant proteins (Fig. 9f). By this method, paxillin was depleted from the cytoplasm, in favor of an enhanced PM localization (Supplementary Fig. 4h), which induced a slower recovery in FAs after photobleaching (fluorescence recovery after photobleaching, FRAP) (Supplementary Fig. 4i, j). The longer half-life and the increased immobile fraction of such PM-anchored paxillin in β3 integrin-containing FAs, together with the upshift of the paxillin band in a WB analysis (Fig. 9g and Supplementary Fig. 1h), effectively demonstrated the occurrence of the cross-linking reaction. The latter, likely prolonging paxillin residency time within these adhesions, rescued the β3[VE/YA] integrin instability (Fig. 9h) and suggested that paxillin stability in adhesions is a key factor in activating the clutch between the substrate-bound integrin and the retrogradely moving F-actin network. Interestingly, strengthening paxillin

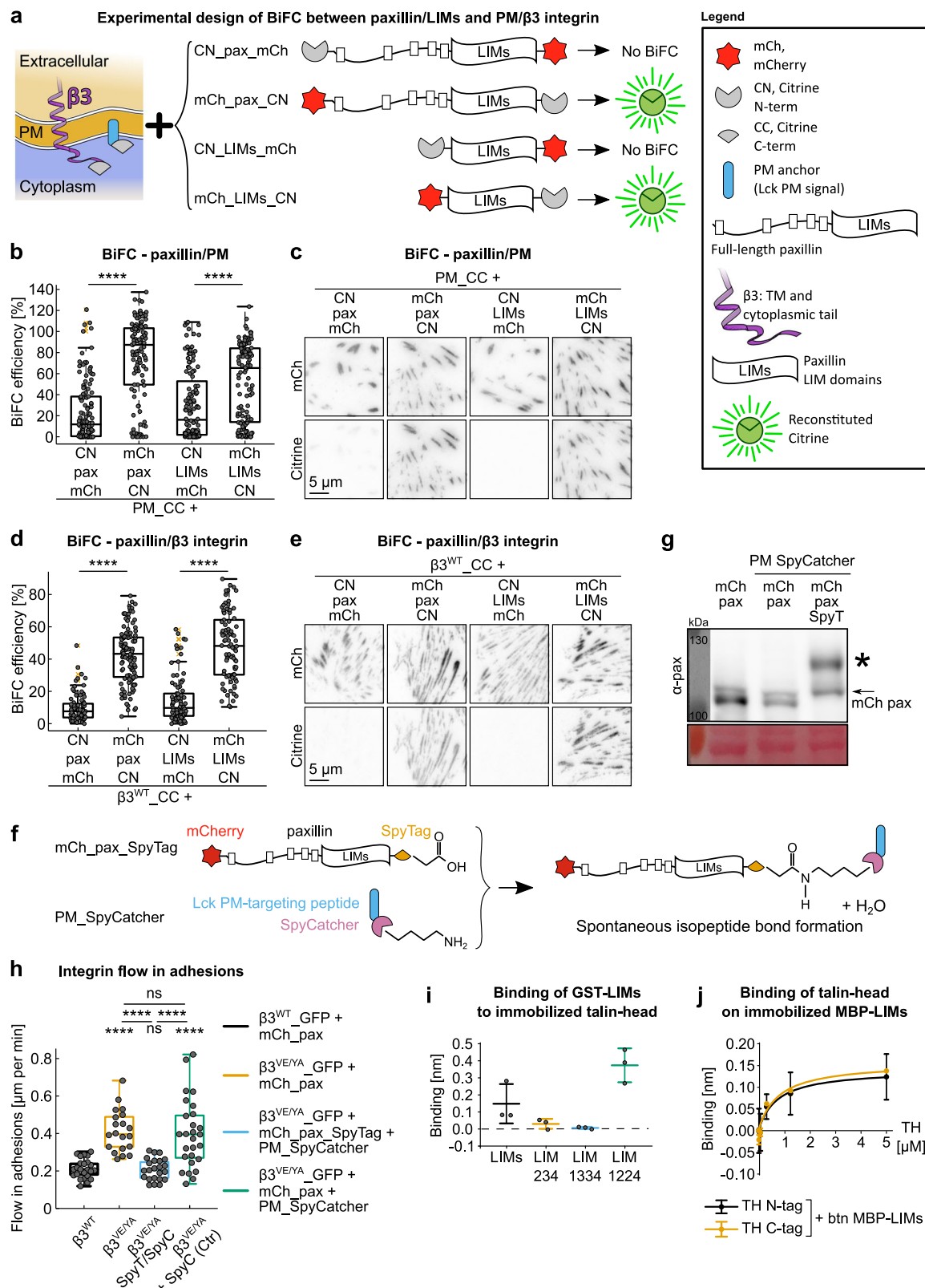

binding to the PM compensate for the loss of paxillin binding to β3 integrin (β3$^{VE/YA}$).

**The paxillin LIM domain interaction network includes talin-head domain binding.** According to the positioning of paxillin within β3 integrin-positive FAs that is emerging from our data, we identified talin-head domain as a putative binding partner of the paxillin LIM domains. Indeed, an interaction of full-length paxillin with talin was recently identified and mapped to the F2, and possibly F3, domains of talin-head by Gao and colleagues, who performed pull-down assays using talin fragments[52]. However, it remains unknown which paxillin domain is involved in this binding to talin-head domain. We therefore performed biosensor experiments with His-tagged talin-head immobilized on

**Fig. 9 Preferential orientation of the paxillin LIM domains within FAs. a** Overview of the molecules tested with the BiFC assays and a summary outcome, in terms of citrine fluorescence, of their pairwise expression. **b, d** Quantification of the BiFC signal generated by the co-expression of each of the paxillin constructs shown in **a** with **b** the plasma membrane-localized CC fragment, in Swiss-3T3 cells or with **d** the β3$^{WT}$ C-terminally tagged with CC, in NIH-3T3 cells. Statistical analysis is provided in Supplementary Dataset 3. **c, e** Representative TIRF images of experiments quantified in **b** and **d. f** Schematic representation of the functioning of the SpyTag/SpyCatcher technology[69] and of how it has been used in our cellular system. **g** Western blot showing the efficiency of the SpyTag/SpyCatcher technology. * indicates the fusion protein originated by the isopeptide bond formation between the mCherry_paxillin_SpyTag and the PM_SpyCatcher. **h** Quantification of the integrins flow in adhesions, expressed as mean displacement over time (μm/min) per replicate. Statistical analysis is provided in Supplementary Dataset 2. **i** Octet biosensor analysis of the interaction between GST-tagged LIM domain variants and Ni-NTA sensor functionalized with His-tagged talin-head domain (talin 1–405 aa). The binding response at 320 nM talin-head domain concentration is reported, $n = 3$ measurements per condition. **j** Octet biosensor analysis of the interaction between MBP-tagged and biotinylated LIM domains (LIMs) immobilized on streptavidin sensor with His-tagged talin-head (TH) domain (talin 1–405 aa), $n = 4$ measurements per condition. ns (not significant) $p > 0.05$; *$p \leq 0.05$; **$p < 0.01$; ***$p < 0.001$; ****$p < 0.0001$. Sample size, complete statistical analysis, and $p$ values are provided in tables in supplementary datasets.

Ni-NTA functionalized biosensor and studied the binding of purified GST-fused paxillin LIM domains (Supplementary Fig. 7c). Free GST was used as a negative control. Both LIM1 deletion (LIM 234) and LIM2 replacement (LIM 1334) abrogated talin binding when compared to the construct containing the original LIM domain-array (LIM 1234), while LIM3 replacement by LIM2 (LIM 1224) potentially augmented this interaction (Fig. 9i). These experiments led us to support the hypothesis of paxillin-talin binding mediated by LIM1 and/or LIM2.

Alternatively, to confirm the specificity of the interaction, we exchanged the fusion partner and the assay layout, and studied binding of His-tagged talin-head on biotinylated MBP-LIMs immobilized on the streptavidin sensors, using free MBP as a negative control (Fig. 9j). To avoid potential problems associated with the tagging of talin, both an N- and a C-tagged version of the protein were tested. Both recombinant talins presented a comparable interaction capacity with the immobilized biotinylated MBP-LIMs with an apparent binding affinity of 720 and 770 nM, respectively (Fig. 9j).

## Discussion

Within the elaborated network of (synergistic) bonds established among FA-proteins, paxillin is among the most connected elements[70]. Together with talin and kindlin, paxillin belongs to the structural subnet that connects integrin receptors to the actin cytoskeleton[70]. However, so far, it has been only described as an adapter or scaffold protein[17]. Nevertheless, the uncoupling of paxillin from the binomial β3 integrin/talin interaction (β3$^{VE/YA}$), produced FAs with structural defects, which could be rescued by the reintroduction of paxillin fragments (Figs. 2a, h and 4). Importantly, our data revealed that the mechanically stable incorporation of paxillin into β3 integrin-containing FAs is distinct from the mechanism of paxillin recruitment to this microenvironment (Fig. 7e, i–l). This stepwise model of paxillin recruitment and docking may involve a transient stabilization by the proposed LIM4-PM interaction[71]. The latter could also account for the paxillin localization to Mn$^{2+}$-induced integrin-kindlin clusters in the absence of talin, which impaired maturation is reminiscent of the unstable β3$^{VE/YA}$ adhesions phenotype[38]. Indeed, paxillin binding to the adhesion complex also reduced the exchange rate of β3 integrins between immature and F-actin coupled integrin-talin clusters[46], as well as the dynamic assembly/disassembly of FAs[54].

The unique combination of molecular and cellular biological tools revealed that the LIM domain-dependent localization of paxillin to β3 integrin-positive FAs[21] is the result of a synergistic binding to FA components (Fig. 6) and possibly the PM (Fig. 8). Although the N-terminal domain of paxillin is not contributing to the targeting mechanism, it further stabilizes paxillin within β3 integrin adhesions, when recruited there by the LIM domain-

array (Fig. 5b–e). This additional stabilization largely relies on phosphorylation of Y$^{31}$ and/or Y$^{118}$ (Fig. 5g, h), which potentially exposes LD motifs, further supporting the cross-linking to FAs proteins, such as talin[16], FAK[21,60], and/or vinculin[10,21,61,62]. Since the N-terminal domain prolonged paxillin half-life over that of talin, it is likely that the paxillin N-terminus interacts with neighboring integrin/talin units, as proposed in the slanted fence model of FAs[1].

On the other hand, the focal adhesion targeting and docking function relies on the paxillin LIM domain-array. The LIM domain replacement strategy (Fig. 7a and Supplementary Fig. 7b) revealed the crucial role of paxillin LIM2 domain for its docking to β3 integrin-positive FAs (Fig. 7e, f). As previously proposed for other LIM domain-proteins[24,48], specific residues confer to distinct LIM domains exclusive functions and determine the selection of precise binding partners. Based on our results, we propose that the deletion or the structural disruption of a LIM domain[21] alters the distance between the remaining domains (Fig. 6a and Supplementary Fig. 7a), undermining multiple synergistic interactions of the LIMs-array with FAs, resulting in a cumulative binding defect. Indeed, the extension of the linker sequence between LIM domains can abrogate a proper protein localization to adhesions[72], suggesting that serial LIM domain modules must have a given relative position to be properly anchored to this complex[31].

Previous results supported an interaction of the paxillin LIM domains with the Y$^{747}$ sidechain, situated in the membrane-proximal NPLY$^{747}$ motif of β3 integrin[30,42–44,49] (Fig. 1a). Although the dissociation rate of paxillin carrying an inactive LIM2 domain (paxillin 1334) was almost exactly that of paxillin wt dissociating from the β3$^{VE/YA}$ integrin-containing adhesions (Figs. 3g, h and 7e, f), we have no structural or biochemical proofs that could indicate an interaction between Y$^{747}$ of β3 integrin and the LIM2 domain of paxillin. Interestingly, the binding of the LIM domains of the Enigma protein to NPLY-like peptides[48,49] was prevented by mutations analogous to our Y$^{747}$A substitution[48]. An alternative binding mechanism of the paxillin LIM1–LIM2-LIM3 array to FAs could involve the recognition of tandem LIM domains by extended peptides of the β3 integrin tail, in a complex similar to that proposed for LIM domain-containing transcription factors[27].

Recently, based on lysine cross-linking experiments, interactions between the N-terminal LD motifs of paxillin and the PH and F0 domains of kindlin-2, as well as between the F0 domain and the paxillin LIM3–LIM4 domains, have been identified and proposed to be responsible for paxillin recruitment to FAs[37]. However, none of these elements was strictly required for paxillin targeting and binding to β3 integrin-containing adhesions (Figs. 6 and 7). Even though the interaction with kindlin might be relevant for paxillin recruitment, it is probably not the major factor

for a stable docking to focal adhesions. Similarly, it remains to be explored to which extent paxillin-talin interactions[16,52], which we showed could also involve paxillin LIM domains (Fig. 9i, j), are pivotal for paxillin recruitment and subsequent binding to β3 integrin-FAs.

Our experiments also propose the LIM4 domain as a candidate for the anchoring of paxillin to the plasma membrane (Fig. 8). Indeed, the local lipid composition of the PM influences the mobility of integrins, clustering and ligand binding[4,46,73], and acts as an anchoring point for indispensable structural FA adapters, such as talin[74,75] and kindlin[76,77]. In addition, transiently recruited paxillin-binding FA-proteins, such as FAK, require membrane interaction and a precise orientation for their activity[78,79]. Likewise, a PM-dependent transient docking could be crucial for paxillin, which was identified as a membrane-proximal protein by proximity biotinylation assays[80]. Moreover, Qin and co-workers recently solved the NMR structure of the kindlin-2 F0/paxillin LIM4 complex[39], which interaction could be part of a kindlin-dependent mechanism of paxillin recruitment to the PM. Interestingly, such an interaction with kindlin does not preclude the binding that we propose between paxillin LIM4 and the PM. Consistent with the hypothesis of a paxillin-PM inter-action, we demonstrated paxillin palmitoylation (Fig. 8e, f). Although the paxillin palmitoylation site(s) could not be clearly identified, the almost exclusive BiFC between the C-terminus of paxillin and the PM-localized probe (Fig. 9b, c and Supplementary Fig. 3c), accredited our hypothesis. Importantly, the paxillin orientation within β3 integrin-FAs relied on the LIMs-array, showing that the precise organization of these domains matters for paxillin docking.

Altogether, our data highlighted the need for comprehensive and integrating studies to understand the mechanisms of paxillin recruitment and adhesion binding to β3 integrin adhesions. Thus, we begin to understand the structural interactions required for the physiological functioning of β3 integrin-positive focal adhesions, both at the level of their assembly, as well as mechanosensing and signal transduction.

## Methods

**Cell culture and transient transfections**. NIH-3T3, Swiss-3T3 fibroblasts, and paxillin-deficient murine fibroblasts were grown at 37 °C (10% CO₂) in DMEM with 4500 mg/L glucose (Sigma-Aldrich), supplemented with 10% heat inactivated FBS (PANBiotech), 1% penicillin-streptomycin (Gibco), and 2 mM glutamine (Sigma-Aldrich). Transient transfections were performed 24 h after seeding cells, with jetPRIME (Polyplus Transfection) according to the manufacturer's recommendations.

Paxillin knock-out cells were generated by CRISPR-Cas9 technology. Three separate guide RNAs were used to target the mouse *paxillin* gene (Santa Cruz Biotechnology, sc-422546). Plasmids expressing gRNA together with Cas9 and GFP were transiently transfected into SV-40 immortalized pre-osteoblast cell line ($\alpha$v$^{f/f}$$\beta$1$^{f/f}$) as previously reported[81]. Single cells were FACSorted based on GFP expression and growing clones were further characterized for the loss of paxillin expression by western-blot and immunofluorescence.

**2-bromopalmitate treatment, acyl-resin assisted capture (Acyl-RAC) assay, and proteomic analysis**. Protein S-acylation was analyzed by the Acyl-RAC assay as described[82], with some modifications. In the case of 2-bromopalmitate treatment, 4 h after transfection, samples were split in two and treated with DMSO or 2-bromopalmitate at 60 µM. Twenty-four hours after transfection, cells were trypsinized, washed with cold PBS and lysed in 300 µl of lysis buffer (1.5% Triton-X100, 25 mM HEPES, 1 mM EDTA in 8 M urea solution supplemented with protease inhibitor cocktail). To block free SH groups with S-methyl methanethiosulfonate (MMTS, Sigma 64306), one volume of 2× blocking buffer (200 mM HEPES, 2 mM EDTA, 5% SDS, and 2.2% MMTS in 8 M urea solution supplemented with protease inhibitor cocktail) was added to the cell lysate and incubated, protected from light, for 5 h at 42 °C in a shaker (600 rpm). Next, proteins were precipitated by adding ice-cold methanol and chloroform (4/4/1: protein mixture/methanol/chloroform), followed by centrifugation at 14,000 rpm for 15 min at 4 °C. The protein pellet was broken using a needle and washed four times with 600 µl of methanol. Air-dried pellet was resuspended in 220 µl of 5× binding buffer (100 mM HEPES, 5 mM EDTA, and 5% SDS in 8 M urea solution supplemented with protease inhibitor

cocktail). Cell debris were removed by centrifugation at 14,000 rpm for 10 min at 4 °C. Five percent of the solubilized pellet was saved as a total cell lysate (TCE) to check for protein expression by western blotting. The remaining was diluted with 8 M urea containing protease inhibitor cocktail, to reach a final volume of 1080 µl, and divided into two tubes. Thioester bonds of one aliquot were cleaved with 600 µl of hydroxylamine (HA, Sigma 55459) freshly dissolved in water to 2.7 M and adjusted to pH 7.4. The control aliquot was treated with equal amount of 2 M Tris-base. Thiopropyl Sepharose® 6B beads (Sigma T8387) were firstly activated for 1 h with mqH₂O, then resuspended in 3 volumes of binding buffer (100 mM HEPES, 1 mM EDTA, and 1% SDS in 8 M urea solution supplemented with protease inhibitor cocktail) and finally added to protein samples (200 µl each). Samples were light-protected and incubated overnight at room temperature on a rotating wheel. Beads were then washed 5× in binding buffer and bound proteins eluted from the beads with SDS-PAGE sample buffer (140 mM SDS, 20% glycerol, 25% β-mercaptoethanol, and bromophenolblue) for 5 min at 95 °C.

Proteomic analysis was performed at the Proteomics Core Facility, Faculty of Medicine, University of Geneva. Samples were in-gel purified and digested with trypsin, alkylated with iodoacetamide, and finally subjected to ESI-LC-MSMS. Peptides were probed against the mouse reference proteome and only those with a threshold higher than 95% maintained for the subsequent analysis. Proteins were identified by a matching of at least two distinct peptides. Non-murine proteins were discarded to avoid contamination by FBS/cell culture reagents and/or sample handling.

**Photoactivation-based experiments (FLAP)**. Image acquisition and analysis were performed at the Bioimaging Core Facility, Faculty of Medicine, University of Geneva. Briefly, transfected NIH-3T3 cells were cultured overnight on coverslips. One hour before imaging, culturing medium was replaced with F12 medium (Sigma-Aldrich), supplemented with FBS, penicillin/streptomycin, and glutamine. Photoactivation was performed on a Nikon A1r confocal laser scanning microscope equipped with a ×60 oil immersion objective and a 37 °C and 5% CO₂ incubation chamber. Laser wavelengths of 488 and 561 nm were used to acquire three pictures at 5 s intervals before photoactivation and 1 frame every 2 s for 3 min after photoactivation. Excitation of photoactivatable GFP molecules was achieved by means of a 405 nm laser (10% power), on a single ROI matching the size of an mCherry-positive focal adhesion. Using Imaris combined with MATLAB scripts, we identified the area of photoactivation, automatically repositioned it in case of lateral shift according to the mCherry signal, and extracted the mean green intensity within the ROI for each time point. The first three time points, corresponding to the background, were averaged and subtracted from the full-time course. The intensity of the first acquisition after photoactivation was set to "100% intensity" and all other values were calculated as ratio. For each photoactivation time series, a constrained double decaying exponentials model (decreasing form) with the formula $\left(\bar{F}_0 - c\right)\left(k_{fast}e^{-\lambda_{fast}\cdot t} + \left(\bar{F}_0 - k_{fast}\right)e^{-\lambda_{slow}\cdot t}\right) + c$ was use to fit the data points. For each of them, the half-life was defined as the time needed to lose 50% of the intensity at time 0, shown in the box and whisker plot, and used for statistical comparison. The same constrained double decaying exponentials model was also used to fit the average data, offering a visual inspection of the goodness of the fit (theoretical model).

**Fluorescence recovery after photobleaching (FRAP)**. FRAP was conducted similarly to photoactivation experiments. However, seeding of transfected cells on coverslips was performed 4 h after transfection to avoid prior irreversible binding of paxillin to the PM via SpyTag/SpyCatcher. Time intervals of acquisition were maintained, while the photoactivation step was replaced by three rounds of scanning using the laser wavelength of 561 nm at 100% power, which efficiently bleached the mCherry fluorophore localized within the selected adhesion. The residual fluorescence was subtracted from the entire time curse, the first three acquisitions, corresponding to the mCherry level prior to bleaching, were averaged and set to 100% intensity, while all other values were calculated as ratio. Individual replicates were fitted with a constrained double exponentials model (increasing form) with the formula $\bar{F}_{fast}\left(1 - e^{-\lambda_{fast}\cdot t}\right) + \bar{F}_{slow}\left(1 - e^{-\lambda_{slow}\cdot t}\right)$. The theoretical half-lives thereof calculated were presented in box and whisker plots and used for statistical comparison.

**Isolation of focal adhesions, immunostaining, and total internal reflection fluorescence (TIRF) microscopy**. After culturing cells overnight on coverslips, cells were either fixed with 4% PFA/PBS or subject to the protocol for the isolation of FAs. In this case, cells were treated as previously described[58] and isolated FAs were fixed immediately with 4% PFA/PBS. For immunostaining, intact cells were firstly permeabilized through a 15 min incubation in PBS containing 1% BSA and 0.5% Triton X-100, while isolated complexes were directly blocked for 30 min with a solution of PBS containing 1% BSA, incubated for 60 min with the primary antibody and subsequently for 45 min with the corresponding secondary antibody, both diluted in the same buffer. Images were acquired on a Nikon Eclipse Ti equipped with a ×100 oil immersion objective. TIRF angle and lasers intensities were kept constant for all experiments. For the quantification of paxillin and 9EG7 within β3_GFP-adhesions or GFP_paxillin-adhesions, as well as for their morphological analysis, these structures were identified with Fiji, applying first the

Phansalkar Auto Local Threshold ($k = 0.1$, $r = 0.1$, and radius = 5) and afterward the Analyze Particle plugin, with the "Exclude on edges" option selected. The mean red and green intensities, the area, and maximal length were then extracted and the number of particles counted. In the case of isolated FAs, for each image acquired, the neurite function of Metamorph was used to identify adhesions on the green channel and subsequently to extract the mean intensity for each of the laser wavelengths used. The quantification of the capacity of cytosolic proteins to be retained in the isolated FAs was expressed in respect to the residual integrin fluorescence, thus as a red/green ratio. In the case of antibody staining, to compensate for possible variations among experiments, the mean red/green ratio in $\beta 3^{WT}$ integrin adhesions was always considered equal to 1, and therefore all the measures expressed with respect to it.

**Bimolecular fluorescence complementation (BiFC).** The citrine fluorophore (FPbase ID VR7EN), generated by the $Q^{69}M$ mutation within the YFP sequence, was obtained as previously described[83]. The N-terminal (CN, aa 1–173) and the C-terminal (CC, aa 174–239) fragments were cloned upstream or downstream the coding sequences of our proteins of interest.

For BiFC assays, transfected cells were seeded on glass coverslips 4 h after transfection and fixed with 4% PFA/PBS 48 h later. Images were acquired on a Nikon Eclipse Ti equipped with a ×100 oil immersion objective. TIRF angle and lasers intensities were kept constant for all experiments. Images were processed with Metamorph, as explained above for isolated FAs, with the only difference that adhesions were identified in virtue of their red fluorescence instead of the green one. For the quantification of the BiFC efficiency, the first step consisted of determining the green background, corresponding to the mean green intensity in mCherry-positive FAs measured by applying the just described analysis to NIH-3T3 and Swiss-3T3 cells only expressing the CN_paxillin_mCherry construct. This value was then subtracted from the mean green intensity in adhesions of each image acquired upon co-expression of pair of BiFC probes. The resulting green value was divided by the red mean intensity in adhesions of the same image. Finally, this ratio was compared to that obtained by expressing solely the corresponding paxillin/LIMs construct presenting the full-length citrine in place of the CN/CC fragment (and analyzed in the same way) considered to be the maximal, 100%, efficiency that could be reached.

**Integrin flow in adhesions.** As for BiFC assays, cells were seeded on coverslips 4 h after transfection. Two days later, the culturing medium was replaced with F12 medium (Sigma-Aldrich), supplemented with FBS, penicillin/streptomycin, and glutamine, and cells were imaged by means of a Nikon A1r confocal laser scanning microscope equipped with a ×60 oil immersion objective and a 37 °C and 5% $CO_2$ incubation chamber. Laser wavelengths of 488 and 561 nm were used to acquire 1 frame every minute for a total of 10 min. Adhesions of a given cell, were segmented frame by frame, using the neurite function of Metamorph applied on the mCherry paxillin or $\beta 3$ integrin-derived fluorescence channel. Only in the case of $\beta 3^{VE/YA}\_CN+mCherry\_paxN\text{-}term\_CC$ and $\beta 3^{VE/YA}\_CC+mCherry\_CN$ adhesions were identified in virtue of the citrine fluorescence because of the non-FAs-specific mCherry localization. Thereafter, Matlab was used to correct the possible translational drifts of cells, and their respective adhesion masks, occurring during the time course of the acquisition. Then background subtraction, followed by a non-local means filter was used to diminish the contribution of spatially stationary areas, i.e., mainly outside of adhesions. Dense optical flow was estimated for all the pixels in the frames, using Farnebäck's method[84]. The velocity of the adhesions flow was computed as the average magnitude of the flow within the adhesion mask between the two following frames.

**Protein extraction and western blotting.** NIH-3T3 transiently transfected were trypsinized and lysed with Ripa buffer (150 mM NaCl, 5 mM EDTA, 25 mM Tris-HCl pH 7.6, 1% Triton X-100, 1% sodium deoxycholate, 0.1% SDS, 1 mM PMSF, and 1 µg/ml chymostatin, leupeptin, antipain, and pepstatin; all obtained from Sigma-Aldrich) during 30 min on ice. Cell lysates were then cleared by a 30 min centrifugation at 14,000 rpm and at 4 °C, supplemented with SDS sample buffer 4× (140 mM SDS, 20% glycerol, 25% β-mercaptoethanol, and bromophenolblue) and boiled for 5 min. Samples were resolved by SDS-PAGE and transferred to nitrocellulose membranes (GE Healthcare). Membranes were blocked in 5% (w/v) BSA (A1391 Applichem) diluted in TBST buffer (0.5 m Tris-base, 1.5 M NaCl, 0.1% Tween 20, pH 7.6), then incubated with primary antibody followed by horseradish peroxidase (HRP)-conjugated secondary antibody, both diluted in 5% BSA in TBST buffer. Staining was revealed using Advansta ECL Quantum (K-12042) or Sirius (K-12043) substrates.

**Octet biosensor analysis and protein biotinylation.** Biosensor analysis was performed on a Fortebio Octet RED384 instrument (Pall Life Sciences) using Ni-nitrilotriacetic (NTA) or streptavidin (SA) sensor by functionalizing of 50 µg/ml His-tagged talin head (talin 1, residues 1–405[85]) or biotinylated MBP-tagged proteins, respectively. Phosphate-buffer (50 mM $NaPO_3$ and 150 mM NaCl, pH 7.2) was used in all steps as running buffer when using the Ni-sensors and the octet kinetic buffer (PBS containing 1% BSA and 0.1% Tween-20, 0.05% sodium azide) was used when using SA-sensors. Every time new sensors were used and prior to the

run, they were soaked into the running buffer for 30 min according to the manufacturer's instructions. The temperature of 27 °C and a stirring speed of 1000 rpm was used throughout the experiments. Ni-sensors were chemically activated by immersing them in 0.05 M 1-ethyl-3-(3-dimethylaminopropyl) carbodiimide (EDC) and 0.1 M N-hydroxysuccinimide (NHS; ThermoFisher Scientific) in Milli-Q water, which enabled covalent coupling of immobilized proteins[85]. 1 M ethanolamine pH 8.5 was used in quenching step for the covalent coupling approach. Biocytin (biotinyl-lysine at 10 µg/ml) was used in quenching step for SA-sensors. The binding of the GST-tagged paxillin constructs with talin head was measured at 20, 80, 320, 1250, and 5000 nM. In the same manner, the binding of the talin head with the biotinylated MBP-tagged paxillin fragments immobilized on the SA-sensor was measured using the same concentrations.

In the case of talin head binding to immobilized MBP-LIMs, streptavidin biosensor functionalized with biotinylated free MBP was used as a reference and the binding response obtained was subtracted from the measured binding between talin and MBP-LIMs to obtain a specific binding response.

In the case of interaction assays between talin head and GST-LIM constructs, binding of free GST on Ni-NTA biosensor-immobilized talin head was used as a reference, which was subtracted from the measured binding between various GST-LIM samples and immobilized talin to obtain a specific binding response.

The MBP-proteins were biotinylated using EZ-Link™ NHS-PEG$_4$-Biotin (Thermo Scientific) according to the instructions of the manufacturer. The non-reacted NHS-PEG$_4$-Biotin was removed by dialysis and the concentration of the proteins was measured by BCA-kit (Thermo Scientific). The biotin incorporation was estimated using HABA (4′-hydroxyazobenzene-2-carboxylic acid; Thermo Scientific) method. The absorbance of the HABA-avidin solution was measured before and after adding the biotinylated sample at 500 nm using Envision UV/VIS (Perkin Elmer). The amount of biotin molecule per sample was then calculated using HABA calculator which is a freely available tool from Thermo Scientific webpage.

**cDNAs and site-directed mutagenesis.** The cDNA encoding the full-length, wild-type and mutant, mouse $\beta 3$ integrins were obtained from the previously described integrin fusion proteins[44]. The linker, coding for the sequence DGSPVAT and containing an AgeI site, was introduced in between the integrin sequence (Uniprot identifier O54890-1) and the fluorescent protein (either GFP, mCherry, PA-GFP, or citrine fragments).

Paxillin fusion proteins were expressed in a cytomegalovirus promoter-driven pcDNA3 vector. All residues of paxillin recombinant proteins were numbered according to the human paxillin α (Uniprot identifier P49023-2). Paxillin N-terminus and LIMs constructs were obtained by the cloning of a PCR-amplified fragment of full-length paxillin, either corresponding to the first 322 amino acids or to the C-terminal sequence starting from residue 317. Paxillin LIM domain deletion mutants were obtained by overlap extension PCR, producing the deletion shown in Supplementary Fig. 5a, and here summarized. Paxillin ΔLIM1: deletion of aa 322–380; paxillin ΔLIM2: deletion of aa 381–439; paxillin ΔLIM3 deletion of aa 441–499; paxillin ΔLIM4 truncation from aa 500. Point mutations were introduced by site-directed mutagenesis. For the LIM domain replacement strategy, we either used serial overlap extension PCRs with degenerated primers or cloned a de novo-synthetized nucleotide sequence, coding for the LIM domain of interest but with a degenerated codons compared to the original one. The amino acid sequence of the final recombinant proteins is shown in Supplementary Fig. 5b. In summary, in paxillin 2234 the aa 324–379 were replaced by the sequence found at aa 383–438; in paxillin 3234 the aa 324–379 were replaced by the sequence found at aa 442–497; in paxillin 1134 the aa 383–438 were replaced by the sequence found at aa 324–379; in paxillin 1334 the aa 382–438 were replaced by the sequence found at aa 441–497; in paxillin 1214 the aa 442–497 were replaced by the sequence found at aa 324–379; and in paxillin 1224 the aa 441–497 were replaced by the sequence found at aa 382–438. The C-terminally tagged paxillins and LIM domains were obtained by removing the stop codon in a PCR step, and then joining the sequence coding for one of the tags via a linker specifying for a GPVAT peptide and containing an AgeI site. GST- and MBP-tagged LIM domain constructs were analogously produced.

The PM-anchored proteins were obtained by sequential cloning of the sequences coding for the PM-targeting peptide of Lck (MGCVCSSNPEL) and the citrine fragment (either CN or CC), or the SpyCatcher, into a pCDNA3 vector. The plasma membrane-targeting sequence was recovered from the Lck-mScarlet-I (98821; Addgene; deposited by D. Gadella[86], a gift from N. Gauthier (IFOM, the FIRC Institute for Molecular Biology, Milan, Italy).

PA-GFP_talin was obtained by replacing the ECFP with the PA-GFP fluorophore, in the chimeric full-length construct previously described[75].

LifeAct_GFP was obtained as previously described[87].

The amino acid compositions of the fluorescent proteins can be found in the FPbase database, at the following IDs. Photoactivatable GFP (PA-GFP): FPbase ID 7QYHY (gift from Christoph Ballestrem, Wellcome Trust Center for Cell-Matrix Research, University of Manchester, Manchester, UK)[88]; mCherry: FPbase ID ZERB6; citrine: FPbase ID VR7EN. SpyTag003/SpyCatcher003 sequences can be found in the Addgene repository[69].

DNA sequence analysis was performed for all constructs and mutants to ensure error-free amplification and correct base replacement.

**Statistics and reproducibility**. Datasets of measurements of the fluorescence ratio did not consistently satisfy normal distribution and therefore statistical comparison was performed using the Wilcoxon test based on ranks. For comparison among more than two groups the Kruskal–Wallis extension was applied, followed by the Dunn's correction. Differences of integrin flows in adhesions among samples were assessed by one-way ANOVA test. The half-lives calculated from FLAP data were subject to one-way ANOVA test, while for half-lives calculated from FRAP experiments the Kruskal–Wallis test was employed. All multiple comparisons were followed by Tukey–Kramer post hoc correction. The nomenclature used in figures is based on the following criteria: ns (not significant) $p > 0.05$; $*p \leq 0.05$; $**p < 0.01$; $***p < 0.001$; $****p < 0.0001$. For each experiment, sample size, complete statistical analysis, and $p$ values are provided in tables in supplementary datasets, as specified in figure captions. All experiments were repeated at least three times and no repeated measurements were performed. For all box and whisker plots presented in this manuscript the box shows the median, 25th and 75th percentiles, while the whiskers the maximum and minimum values. Reported outliers are indicated with the symbol $x$. FLAP data are shown as mean intensity ± standard error of the mean (experimental) or constrained double decaying exponentials model ± standard error of the mean (theoretical dissociation model).

**Reagents**. Purified mouse monoclonal anti-paxillin (BD Biosciences 610051, 1:1000 dilution). Purified Rat Anti-Mouse CD29, clone 9EG7 (BD Pharmingen™, 1:600 dilution). Mouse monoclonal anti-vinculin (Sigma-Aldrich V9131, 1:600 dilution). Rabbit polyclonal anti-caveolin-1 (Santa Cruz Biotechnology sc-894, 1:4000 dilution). Rabbit polyclonal anti-mCherry (BioVision 5993-30 T, 1:4000 dilution). Alexa Fluor 555 anti-mouse (ThermoFisher A-31570, 1:500 dilution). Alexa Fluor 633 anti-mouse (ThermoFisher A-21050, 1:500 dilution). Mouse monoclonal anti-GFP, clone B34 (BioLegend, 1:50,000 dilution). HRP-conjugated anti-mouse (Jackson, 1:10,000 dilution). HRP-conjugated anti-rabbit (Jackson, 1:10,000 dilution).

**Reporting summary**. Further information on research design is available in the Nature Research Reporting Summary linked to this article.

## Data availability

Microscopy images that support the findings acquired during the current study are available in Yareta with the identifier doi: 10.26037/yareta:23jacb27ibdm5erlqxcdluzwrm[89]. The mass spectrometry proteomics data have been deposited to the ProteomeXchange Consortium (http://proteomecentral.proteomexchange.org) via the PRIDE partner repository[90] with the dataset identifier PXD023810 and DOI 10.6019/PXD023810[91], while detailed sample handling by the Proteomics Core Facility is available in Yareta. List of figures that have associated raw data: 1b–n, 2a–b, d–h, 3a–i, 4a–h, 5b–e, g–i, 6c–g, 7b–l, 8b–h, 9b–e, g–j, Supplementary 1a–h, 2a–e, 3a–d, 4a, c–f, h–j, 5a–l, 6a–d, 8a–i. Source Data for graphs are available in Supplementary Dataset 6. Any other remaining information is available from the corresponding author upon reasonable request.

## Code availability

MATLAB codes were developed in-house at the Bioimaging Core Facility, Faculty of Medicine, University of Geneva. The MATLAB code used for automatically reposition the photoactivated ROI in case of lateral shift during the acquisition time is available at https://github.com/Bioimaging/XTBxyt_ShapeTracker. The MATLAB code used to estimate the dense optical flow for all the pixels in the frames, using Farnebäck's method, is available at https://github.com/Bioimaging/Sliding-adhesions/tree/main.

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

## Acknowledgements

We are grateful to Dr. Sunil Dogga for introducing us to the Acyl-RAC technique, to Dr. Patricia Vazquez for her contribution in the initial phase of this work and to Monica Julio Barreto for the technical help. All the members of the Bioimaging Core Facility of the Center Médical Universitaire are acknowledged for help in microscopy and image analysis. We thank all the members of the Proteomics Core Facility of the Faculty of Medicine, University of Geneva, for their engagement in the challenging task of processing our samples. This work was supported by the Swiss National Science Foundation to BWH, grants 31003A_166384 and 310030_185261 and by Academy of Finland to VPH, grant 331946. We also acknowledge Biocenter Finland for technical support.

## Author contributions

M.R. and B.W.H. designed the DNA constructs and experiments. M.R. conducted and analyzed the experiments. N.L. designed the strategy to analyze the photoactivation and FRAP data and the proteins flow in adhesions. L.A. and V.P.H. performed the interaction analyses between talin and paxillin and supported the design of SpyTag/SpyCatcher constructs. D.B. generated and provided the paxillin-null cells. M.R. and B.W.H. wrote the manuscript. All authors accepted the final version of the manuscript.

## Competing interests

The authors declare no competing interests.
