## [Peer Review File · Communications Biology]

Reviewers' comments:

Reviewer #1 (Remarks to the Author):

This is an interesting study that revisits earlier studies by various labs regarding the role of paxillin in focal adhesion dynamics and the respective roles of paxillin subdomains in its function and focal adhesion targeting. The authors use a wide array of fluorescently-tagged chimeric protein tools including chimeric $\beta 3$ integrins, previously shown to exhibit modulated talin and/or paxillin association in cells, BIFC constructs to drive artificial protein-protein interactions, PA-GFP, and others to drive plasma membrane lipid association. Critical roles for the LIM domains in FA targeting are confirmed, as are the supporting roles for the LD motifs and Y phosphorylation sites in the amino terminus. Novel observations include evidence that paxillin (LIM domain) recruitment is important in stabilizing integrin $\beta 3$ adhesions, that the spacing and orientation of the paxillin LIM domains are key to its function, perhaps more importantly than primary sequence and that the LIM4 domain of paxillin may bind phospholipids in the PM to facilitate this orientation. Despite these findings, the study falls short of providing critical new insight into what the paxillin LIM domains actually interact with in focal adhesions to elicit these functions.

Specific points

1. The study is technically well executed but has some major limitations given that all of the studies involve overexpression of the various chimeric proteins in a background of endogenous paxillin and $\beta 3$ integrin. Indeed, as noted by the authors (eg. line 124), this could influence interpretation of their data. Thus, it is recommended that some of the key findings are repeated in a paxillin deficient background.
2. Since the focus of this study is on $\beta 3$ integrin, the authors should be careful to acknowledge that some of their findings, including the role of specific LIM domains in paxillin targeting may not hold for other integrins eg $\alpha 5\beta 1$ /or cell types. Therefore, they may want to either test other cell types/integrins or tone down their conclusion and perhaps specify $\beta 3$ integrin adhesions in the title.
3. While PA-GFP tagging paxillin on either N or C term was deemed to have no impact on binding FAs (Supp 3a-c) the authors present evidence that LIM4 is critical for PM association, palmitoylation and FA orientation and function of paxillin. Since tagging LIM with a bulky fluorophore may be expected to interfere sterically, this apparent discrepancy should be discussed.
4. The adhesions highlighted in cells expressing $\beta 3\text{VE}$ (Fig 1) look more like those of the $\beta 3\text{VE}/\text{YA}$ expressing cells than WT $\beta 3$. Movies of all 3 should be included for comparison. In that regard, since the adhesions don't disassemble can the authors exclude the possibility that "sliding" is not an indirect consequence of either cell rounding or enhanced contractility?

Reviewer #2 (Remarks to the Author):

The manuscript by Marta Ripamonti et al provide a mechanistic insight into the contribution of individual Paxillin LIM domains in targeting and maintaining the structural integrity of FA. While the contribution of paxillin LIM domains acting as scaffolds to target multiprotein complexes to FA sites is known, this work demonstrates that precise spacing of LIM domains within the sequence possibly act as a mechanical tension determinant. Although the idea of proper spacing between LIM domains affecting localization to FAs is previously suggested (Petit, M.M. et al., 2003), the current study has confirmed this experimentally.

The authors additionally claim that both LIM1 & 2 are comparable to LIM3 for FAs targeting with LIM2 being the major determinant in contrast to LIM3 as reported earlier (Brown et al 1996). Each LIM1,2,3 have unique functional and spatial signatures and are not interchangeable.

Further, authors claim an additional function of Paxillin N-terminus and highlight the importance of tyr31 and 118, phosphorylation sites for FAK and vinculin binding in stabilizing LIM domain dependent paxillin localization at FAs which is in contrary to previous findings (Brown et al., 1996).

Additionally, the author highlighted the role of Paxillin LIM4 Cys512 and Cys557 as potential palmitoylation sites that aid in association with PM and integrins and thus influence the maturation of integrin clustering and association with its adaptor proteins.

The above-mentioned claims although not entirely novel in conception as most of these findings are inferred from previous studies but has been systematically, experimentally confirmed in this study for the first time using convincing molecular and cellular assays. Overall, this work provides a comprehensive understanding of physiological functioning of FAs and how structural interaction and recruitment of paxillin influence the FAs integrity.

To this reviewer the manuscript seems publishable when the authors address the following minor concerns.

♣ Compromised paxillin binding in signaling defective integrin β 3VE/YA as demonstrated in (Fig1b, g) is not visually convincing. The representative images show substantial paxillin recruitment and association with β 3VE/YA. I would suggest doing a colocalization quantification (Mander's coeff.) for paxillin and beta integrin WT & mutants to confirm any substantial difference in paxillin recruitment. The instability of FAs is rather indicative of rapid dissociation of paxillin from β 3VE/YA adhesion sites as addressed in the next section rather than compromised paxillin binding.

♣ Live cell imaging movie for β 3VE is missing (line 108). The Supplementary Movies 1 and 2 are for β 3WT and β 3VE/YA.

♣ Figure7d show no significant effect of 2BP treatment on the palmitoylation levels in cell lysates expressing WT mCh Paxillin between control and treated samples. Authors could provide a better representative pulldown image for the Acyl-Resin Capture assay.

Reviewer #3 (Remarks to the Author):

Ripamonti et al., Structural and functional analysis of LIM domain-dependent 1 recruitment of paxillin to focal adhesions

In the present study, Ripamonti et al analyze integrin-dependent mechanisms of the LD and LIM domain-containing protein paxillin recruitment of focal adhesions (FAs) using a set of elegant cell biological and imaging techniques. They show that mechanical coupling of paxillin in the FA to the plasma membrane or integrin is important for FA stability and integrin-talin linkage. Uncoupling paxillin using a mutant β 3 integrin in a putative paxillin binding site led to increased intgrin flow in FAs and reduced mechanical stability of FAs. The authors went on to define the contribution of the individual LIM domains for FA localization and function using a series of paxillin variants (deletion and domain swapping) and demonstrate the spatial orientation of the paxillin LIM domains, with LIM4 closest to the plasma membrane.

In general, the manuscript contains a lot of information, is well written and structured. Although some results are confirmatory the study provides sufficient amount of novel results (contribution of individual LIM domains and the orientation of the paxillin LIM domains) to be of great interest for the community. However, I am not entirely convinced that the experimental setup supports all the conclusions. The imaging techniques including BiFC, the focal adhesion isolation, FA sliding are very elegant but one wonders if some conclusions, e.g. plasma membrane interaction of LIM4, would require additional experiments to draw these conclusions. Also, the presence of endogenous paxillin might mask functions of mutant paxillin variants when co-expressed in the same cell. Additional experiments are necessary to draw the conclusions and to support publication of the study in Communications Biology.

Major points:

1. One point of criticism is the expression of proteins in the background of the endogenous proteins. While this might not be problematic in case of the $\beta 3$ integrin variants as NIH-3T3 cells express low levels of endogenous $\beta 3$ integrin, the analysis of paxillin variants could be more problematic. We have observed in the past and studies have shown that GFP-tagged proteins might not be able to localize properly in the presence of the endogenous protein but does so in the absence of the endogenous protein. It would have been a clearer setup to study paxillin in paxillin-depleted NIH-3T3 cells. Although this might be too much for all experiments it should be mentioned in the results or discussion section.

2. What is the effect of a paxillin knockout or knockdown on the FAs sliding phenotype, FA stability etc? Would the authors expect a similar phenotype to the $\beta 3$ VE/YA integrin? This would illustrate how much of paxillin function in FAs is attributed to its localization within FAs and to its proper mechanical linkage to the membrane/integrins.

3. The authors rescued the $\beta 3$ VE/YA integrin the FA sliding phenotype by BiFC-based mechanical coupling of paxillin to $\beta 3$ VE/YA integrin and suggest that a compromised paxillin binding and retention within FAs is the underlying cause of the sliding phenotype. Was it analyzed if the BiFC-based method indeed led to increased mechanical coupling of paxillin to $\beta 3$ VE/YA integrin e.g. upon isolation of FA complexes (done in Figure 3). Did the authors examine the expression levels of mCherry-paxillin vs mCherry-paxillin-CN (or CC) to rule out that dissimilar protein levels cause the observed effects?

4. How do the authors explain the reduced half-life of paxillin in $\beta 3$ VE integrin adhesions despite comparable integrin flow in adhesions? Isn't this contradicting the authors hypothesis that paxillin stability in FAs and its coupling to integrins is the underlying cause of the sliding phenotype?

5. Figure 4: It is surprising that LIM-only rescue the adhesion flow while the N-terminal part with the two tyrosine residues contribute the paxillin stability in FAs (figure 4). Did the authors use the paxillin N-terminal half for the BiFC experiments in figure 4i or any other unrelated citrine-tagged protein as negative control to rule out that any citrine-tagged protein would slow down the flow in adhesions?

6. Figure 6,7: Did any of the mutant (LIM domain replacement mutants or point mutant in the LIM4 domain) affect localization of the respective paxillin construct to FAs? Perhaps the authors could include the TIRF images of the different variants as supplementary figure.

7. Figure 7: The authors write that they used "site-directed mutagenesis, preventing palmitoylation at Cys512/557 or abrogating LIM4 positive charge, did not affect paxillin expression levels" (line 391f). However, they did not test if the cysteine mutations indeed affect palmitoylation. This should be included and would be a perfect control to show that their Acyl-Resin Assisted Capture assay indeed shows a small but specific pool of palmitoylated paxillin. (the authors do write in the discussion that one would need to screen 24 cysteines but since they only write about palmitoylation in the context of the LIM4 domain the results part, it should be tested)

8. Figure 8: The mutational analysis of paxillin would benefit if some of the mutant e.g. Cys512/557 or abrogating LIM4 positive charge would be included in the BiFC analysis (Figure 8b,d) to determine if any of these mutations indeed affect the proper orientation of paxillin to the plasma membrane or $\beta 3$ integrin.

Minor points:

1. It would be beneficial for readers if a brief description of the NIH-3T3 cells could be included; specifically, that these cells express low levels of $\beta 3$ integrin endogenously (this is mentioned in a previous publication by the Wehrle-Haller lab but could be mentioned here again). While this explains that the phenotype of transiently-expressed mutant $\beta 3$ integrins is prominent I was puzzled by the fact that $\beta 1$ integrins do not seem to play a major role for mCherry-paxillin localization (no paxillin in $\beta 1$ integrin containing fibrillar adhesion) or FA displacement despite the fact that cells were grown in full medium (which should contain FN and VN) and without special coating of surfaces with VN.

2. If possible, the authors should quantify the FA instability and accelerated retrograde sliding shown in the supplementary movies 1+2.

3. Line 213f: The authors write "Indeed, while in intact cells, paxillin was recruited to β 3WT and β 3VE adhesions equally well (Fig. 1b), in isolated FAs, the...". However, the paxillin recruitment to β 3WT and β 3VE adhesions was not quantified and only analyzed qualitatively. The authors would need to measure fluorescence intensities and determine their ratios (β 3 vs paxillin) to make such a statement.

4. The authors write that their study shows that the LIM4 domain interacts with the plasma membrane (e.g. line 88, 429). However, this seems an overstatement as they only show proximity between the LIM4 and the plasma membrane but not its interaction with lipids (they also did not define the location of the potential palmitoylation).

5. Please include the reference for the Acyl-Resin Assisted Capture assay. Gadalla, Abrami and Goot is written in the methods section but not listed in the references.

Dear Reviewers,

We would like to thank you for handling and evaluating our manuscript, as well as for suggesting us additional interesting experiments. You will find here an overview of the major elements which we modified and the new results that were added to the figures and text to make the manuscript suitable for publication, followed by our responses to the specific concerns that you raised.

All three reviewers felt that we should discuss and experimentally address a potential interference of endogenous paxillin and that of other endogenous integrins on our experimental data. To address these issues, we performed new experiments in cell lines presenting different expression levels of paxillin and $\beta 3$ integrin. These included the NIH-3T3 (expressing very low levels of endogenous $\beta 3$ integrin), as well as a previously described clone of 3T3 cells expressing endogenous $\beta 3$ integrin (Swiss-3T3), and in addition, paxillin-null fibroblasts, as requested by the reviewers. In particular, we evaluated and quantified the paxillin/integrin ratio within adhesions, as this might be influenced by an endogenous pool of competing proteins. Thus, the mCherry_paxillin was co-expressed with $\beta 3$ _GFP integrins in both NIH-3T3 and paxillin-null cells, and we compared the resulting staining ratio in respect to endogenous paxillin alone (Fig. 1). We also ensured that paxillin recruitment to adhesions containing mutant $\beta 3$ integrins, were not influenced by endogenous $\beta 1$ integrin receptors and that adhesions morphology was not influenced by paxillin expression levels (Supp. Fig. 1 and Supp. Fig. 2). Furthermore, we tested the effect of paxillin knock-out on the dissociation kinetics of two different paxillin constructs, namely paxillin wild-type and paxillin 1334 (Supp. Fig. 7). Since the dissociation of photoactivated paxillin was among the most sensitive tests to determine paxillin binding defects, we are confident that the data present in our manuscript are not influenced by endogenous wild-type paxillin levels as we were not able to show any significant difference in respect to the cell line used (NIH-3T3 *versus* paxillin-null). This indeed suggests to us that the dissociation dynamics of the exogenously expressed paxillin constructs is not significantly influenced by the presence of endogenous paxillin. On the other hand, when applying the co-localization test in focal adhesions, we revealed that a competition between wild-type full-length GFP-tagged paxillin and various mCherry-tagged paxillin mutants leads to a lower accumulation of mutant paxillins within adhesions (Supp. Fig. 7). In fact, these observations have now been added to the manuscript and they reveal that competition among paxillin isoforms can affect the steady state conditions of paxillin localization to focal adhesions, but is not revealing the binding interaction effectively occurring within focal adhesions. Since the dissociation kinetics can also be precisely evaluated with a smaller pool of recruited paxillin, it is plausible that its dissociation kinetics is regulated by the actual binding interaction and not the concentration of potentially competing paxillin proteins in the same adhesion.

Besides including some control conditions in the test of $\beta 3^{VE/YA}$ integrin-induced adhesions sliding, we tried to shed further light on the underlying cause of this interesting phenotype. In addition, we also asked whether this integrin sliding was correlated with the flow of the actin cytoskeleton. In fact, the high-affinity talin-binding site led to very similar retrograde sliding parameters between the F-actin network and the embedded integrins (Fig. 2). Since these data indicated a potential inside-out signaling defect in the VE/YA mutant, we further investigated whether the introduction of an integrin activation mutation in the integrin ectodomain (N^{305T}) could rescue the integrin sliding. Indeed with this newly generated integrin mutant ($\beta 3^{N305T+VE/YA}$) we could demonstrate the halt of the inward sliding of integrin receptors, despite the lower affinity to paxillin. These new data are also in line with a new requested control experiments, showing that rescuing of the sliding was achieved with BIFC-dependent coupling of the paxillin N-terminus (LD motifs-containing) to the $\beta 3^{VE/YA}$ integrin.

Further significant additions to the manuscript consists in the proteomic analysis of palmitoylated proteins precipitated with the Acyl-RAC experiments (Fig. 8). Although we were not able to identify the palmitoylated

cysteine with this strategy, we were able to validate our experimental strategy as 95% of the precipitated proteins have been previously identified as palmitoylated and annotated in the SwissPalm database.

In order to better understand how the paxillin LIM domains are oriented and bound within focal adhesions, we have also added protein-protein interaction experiments, based on the Octet biosensor, showing that the paxillin LIM domains interact with the talin-head domain, therefore proposing a new binding interaction of the LIM1-LIM2 module of paxillin on the integrin-binding talin-head domain (Fig. 9).

Below you find our responses, point-by-point, to your comments.

With best regards,

Marta Ripamonti and Bernhard Wehrle-Haller

Reviewer #1

- This is an interesting study that revisits earlier studies by various labs regarding the role of paxillin in focal adhesion dynamics and the respective roles of paxillin subdomains in its function and focal adhesion targeting. The authors use a wide array of fluorescently-tagged chimeric protein tools including chimeric $\beta 3$ integrins, previously shown to exhibit modulated talin and/or paxillin association in cells, BIFC constructs to drive artificial protein-protein interactions, PA-GFP, and others to drive plasma membrane lipid association. Critical roles for the LIM domains in FA targeting are confirmed, as are the supporting roles for the LD motifs and Y phosphorylation sites in the amino terminus. Novel observations include evidence that paxillin (LIM domain) recruitment is important in stabilizing integrin $\beta 3$ adhesions, that the spacing and orientation of the paxillin LIM domains are key to its function, perhaps more importantly than primary sequence and that the LIM4 domain of paxillin may bind phospholipids in the PM to facilitate this orientation. Despite these findings, the study falls short of providing critical new insight into what the paxillin LIM domains actually interact with in focal adhesions to elicit these functions.

We thank the reviewer for this critical evaluation of our manuscript, and agree that we have not yet been able to identify the different interacting partners of the paxillin LIM domains in focal adhesions. However, we want to stress here that previous attempts to perform such kind of studies left the field in confusion. In fact, the real relevance of some of the paxillin LIM domains and their previously claimed roles in paxillin recruitment need to be reconsidered. The lack of clear information in the field has led us to undertake the current study. For example, we now show that simple LIM domain deletion strategies associated with a focal adhesion co-localization approach will not allow to properly determine paxillin binding to these complex structures and the specificities of the LIM domain interactions. On the contrary, the techniques that we have introduced in this manuscript provide the necessary specificity to distinguish between a loose paxillin association with focal adhesions and its firm docking into the complex and could be employed for the study of other relevant protein-protein interactions within integrin-dependent cell-matrix adhesions.

Nevertheless, we have attempted in this revision to introduce additional data that should point to the possibility of direct protein-protein interactions of the most N-terminal LIM domains (LIM1 and LIM2) with the focal adhesion adapter protein talin. This biochemical interaction is in line with the recently published FERM-folded structure of talin (Zhang et al. 2020), as well as some biotin-ligation data showing paxillin interaction with the F1-domain of the talin-head, positioned in the immediate proximity of the membrane proximal NPXY-binding site of the integrin tail. Together with the putative lipid interaction module located in the C-terminal LIM domain of paxillin (LIM4), we provide now additional structural insights into the way paxillin is interacting with $\beta 3$ integrin, talin and membrane to fulfill its structural and signaling roles in $\beta 3$ integrin-positive focal adhesions.

1. The study is technically well executed but has some major limitations given that all of the studies involve overexpression of the various chimeric proteins in a background of endogenous paxillin and $\beta 3$ integrin. Indeed, as noted by the authors (eg. line 124), this could influence interpretation of their data. Thus, it is recommended that some of the key findings are repeated in a paxillin deficient background.

We thank the reviewer for the positive evaluation of the technical qualities of our study. We also agree that some of the mutational studies would benefit from a paxillin-null background, in order to exclude interference by the endogenous protein. While a knockout study with additional mutagenesis may be a relevant approach to dissect mono-specific protein-protein or protein-lipid interactions, such an approach is almost impossible to conduct when the protein is forming multiple different interactions

within a protein-interaction network. Thus, removing one of such a critical component may destroy the entire network and many physiological associations can no longer be investigated because the entire focal adhesion structure can no longer be formed by the cell. In order to discuss this issue and to provide to the reader a set of data that allow to judge the role of endogenous paxillin, we have conducted a number of control experiments with paxillin-deficient mouse fibroblasts. In addition, we also co-expressed wildtype and mutant paxillin constructs in the same cells in order to judge the potential competition of an endogenous wild-type protein with a transfected mutant paxillin construct. These data show that the amount of paxillin expressed in cells matters when analyzing paxillin recruitment mechanisms that are linked to potentially multiple low-affinity interactions. However, more specific approaches, such as the analysis of dissociation kinetics are needed to better understand the few high-affinity docking sites within $\beta 3$ integrin-adhesions. Interestingly, while the former mechanisms appear to depend on the paxillin expression levels in the cell (Supp. Fig. 7d-g vs Supp. Fig. 7h,i), the more specific interactions measured by dissociation kinetics appear not to be altered by the presence of endogenous paxillin (Supp. Fig. 7a-c).

In the revised manuscript we have therefore included a series of experiments which helped to determine whether endogenous paxillin has influenced the interpretation of our data:

- Fig. 1d,f
- Fig. 1h-m
- Supp. Fig. 2a-c
- Supp. Fig. 7

Altogether our data showed that the localization of recombinant paxillin variants to $\beta 3$ integrin-positive adhesions is not influenced by the presence of endogenous paxillin, likely because of the relative low level of expression of the latter. However, a competition in favor of the wild-type isoform over the mutants can be shown by the concomitant overexpression of differentially tagged paxillins (GFP and mCherry, Supp. Fig. 7h,i), and potentially restricting the availability of $\beta 3$ integrins (not overexpressed but endogenously expressed by Swiss-3T3 cells).

Concerning the photoactivation experiments and the interpretation of paxillin dissociation kinetics from $\beta 3$ -positive FAs, we have performed the same experiment and measured the FLAP for paxillin wt and paxillin 1334 in the paxillin-null cell line (Supp. Fig. 7a-c). No differences could be detected between these proteins in respect to the two cell lines used, suggesting that the binding capacity of paxillin to $\beta 3$ integrin-adhesions is an intrinsic property of each paxillin mutant analyzed.

Regarding a possible role of other integrins in the recruitment of paxillin to adhesions presenting $\beta 3$ integrin mutants, we have investigated the activation of $\beta 1$ integrin receptors and excluded a compensatory function based on similar levels of 9EG7 staining within $\beta 3^{\text{WT}}$ and $\beta 3^{\text{VE/YA}}$ integrin-adhesions (Supp. Fig. 1).

2. Since the focus of this study is on $\beta 3$ integrin, the authors should be careful to acknowledge that some of their findings, including the role of specific LIM domains in paxillin targeting may not hold for other integrins eg $\alpha 5\beta 1$ /or cell types. Therefore, they may want to either test other cell types/integrins or tone down their conclusion and perhaps specify $\beta 3$ integrin adhesions in the title.

The title, initially generally referring to FAs, has been changed in order to correctly refer to the type of FAs that we have been investigating, meaning $\alpha \beta 3$ -positive focal adhesions. Moreover, a better explanation regarding the formation of $\beta 3$ integrin-containing heterodimers in NIH-3T3 cells has been included, as suggested by Reviewer #3 as well.

Being aware of the specificity of this study to the analysis of $\beta 3$ integrin-positive adhesions, we have been trying to always specify that the FAs investigated had this characteristic of expressing the $\beta 3$ integrin. We have now corrected the text, in order to avoid the generalization of our findings to all types of integrin-dependent adhesions.

3. While PA-GFP tagging paxillin on either N or C term was deemed to have no impact on binding FAs (Supp 3a-c) the authors present evidence that LIM4 is critical for PM association, palmitoylation and FA orientation and function of paxillin. Since tagging LIM with a bulky Fluorophore may be expected to interfere sterically, this apparent discrepancy should be discussed.

We thank the reviewer for highlighting this relevant point, as we may have not paid enough attention to explain this interesting difference. As indicated above, we think that paxillin binding to $\beta 3$ integrin-adhesions relies on the interaction with multiple different partners. While LIM domains contribute the most to the recruitment of paxillin to $\beta 3$ integrin-adhesions, the N-terminal LD motifs are contributing to almost half of the paxillin retention once recruited in adhesions by the LIM domains. Therefore, the context of the paxillin interaction with focal adhesions is dramatically changing once the LD motifs get entangled with other focal adhesion components, such as vinculin and/or the rod domain of talin. Within this more complex environment, a small potentially steric disadvantage of a C-terminal tag is compensated by the entire interaction network of the full-length paxillin (i.e. PA-GFP_paxillin and paxillin_PA-GFP have the same dissociation kinetics). In contrast, and due to the very precise analysis of the dissociation kinetics, a small steric interference was noticed when we analyzed the LIM domains alone (i.e. PA-GFP_LIMs vs. LIMs_PA-GFP). Based on the dynamic measurements of fluorescently tagged full-length paxillin, which was linked to the fluorophore either at the N- or C-terminus, we would have concluded that, despite a potential function of LIM4 to interact with the PM, the presence of the bulky fluorescent protein at the C-terminus had only a negligible influence on the dissociation kinetics. However, when the LIM domains were subject to the same experiment (Supp. Fig. 4a-c), a significant difference was detected based on the positioning of the tag, suggesting that indeed the tag at the C-terminus is slightly reducing the interaction affinities with focal adhesions.

In the context of the N-terminally-tagged full-length paxillin, deletion and point mutations in the LIM4 domain reduced the $t_{1/2}$ of paxillin in FAs by half, being roughly equal to the effect caused by the deletion of the entire paxillin N-terminus (LIMs). Based on these relative differences, we can conclude that the potential steric interference at the C-terminus of paxillin caused by the fusion of a fluorescent tag is much lower than that obtained with the removal or mutation of the LIM4 domain, which proposes that the results obtained with the C-terminally-positioned BiFC probes are trustworthy. In fact, BiFC between full-length paxillin/LIMs and PM were possible when the tag was at the C-terminus, suggesting that it has not sterically prevented the juxtaposition of LIM4 to the PM (Fig. 9b,c; Supp. Fig. 3c). In other words, the tagging was required for our experiments, however the combination of tagging positions and the diversification of analyses performed (e.g. effective BiFC within $\beta 3$ integrin-adhesions) suggest that the fluorophore/fluorophore fragments did not severely compromise the set of paxillin interactions within $\beta 3$ integrin-FAs. On the contrary, if we force an interaction between the N-terminus of LIMs and the plasma membrane (Supp. Fig. 3f), the association of paxillin with focal adhesions is altered, likely due to changes in its specific orientation causing alterations to the interaction network.

4. The adhesions highlighted in cells expressing b3VE (Fig 1) look more like those of the b3VE/YA expressing cells than WT b3. Movies of all 3 should be included for comparison. In that regard, since the adhesions

don't disassembly can the authors exclude the possibility that "sliding" is not an indirect consequence of either cell rounding or enhanced contractility?

We were also surprised to see the differences in adhesion behavior as well as adhesion form between the different integrin mutants. To make the comparison easier for the readers, we have included additional data in order to allow better appreciation of the differences between VE and VE/YA mutations. Accordingly, a quantitative analysis of the morphology of $\beta 3_{\text{GFP}}$ FAs was carried out, assessing the average area and the maximal length of adhesions per acquired image (typically one image contains one cell and potentially some additional adhesions from neighboring cells) (Fig. 1h-m; Supp. Fig. 2a-c). $\beta 3^{\text{VE/YA}}_{\text{GFP}}$ FAs appeared bigger and longer than that formed in cells transfected with the wild-type or the VE-variant, in the two cell lines tested (NIH-3T3 and paxillin-null) and irrespective of paxillin overexpression. On the other hand, an increased number of $\beta 3^{\text{VE}}_{\text{GFP}}$ FAs was detected per image.

A movie of $\beta 3^{\text{VE}}_{\text{GFP}}$ adhesions has also been included as Movie 2 (while the previous Movies 2 and 3 have been re-numbered as Movies 3 and 4). As indicated by the quantification, focal adhesions consisting of $\beta 3^{\text{VE}}_{\text{GFP}}$ are only slowly sliding, similarly to that containing $\beta 3^{\text{WT}}_{\text{GFP}}$.

Although the cell area was not quantified, cells expressing $\beta 3^{\text{VE/YA}}$ integrins (either GFP- or mCherry-tagged) appeared flat and did not round-up during the time course of the experiment. Similarly, we did not detect rounding of $\beta 3^{\text{VE/YA}}$ -expressing cells in culture or when seeded on coverslips and were not detected after fixation either, suggesting that cells were able to properly attach under the given, serum-containing, culturing conditions. These observation did not suggest that cells were more contractile or would retract more easily when imaged under the microscope. In order to possibly provide an explanation for the observed sliding phenotype we have determined the actin flow in $\beta 3$ integrin-adhesions (Fig. 2b) and showed that when the $\beta 3$ integrin presents the high-affinity talin binding site, the flow of the integrin and that of actin were comparable. Moreover, these measures of the actin retrograde flow in $\beta 3^{\text{WT}}$ and $\beta 3^{\text{VE}}$ adhesions are consistent with the F-actin flow in the lamellum determined in the work of Waterman and colleagues (Thievessen et al. 2013). On the contrary, the actin flow within $\beta 3^{\text{VE/YA}}$ adhesions appeared equivalent to that measured in nascent adhesions (Thievessen et al. 2013). According to these observations we have reasoned that the VE modification would couple the integrin receptor to the F-actin network through the stronger talin interaction, and that this complex would move together with the actin flow. Accordingly, a stable integrin/extracellular matrix connection (or an engaged clutch) should slow the retrograde sliding. In line with this hypothesis, the sliding of the $\beta 3^{\text{VE/YA}}$ integrin could be rescued by combining this mutation with the dominantly activating mutation N³⁰⁵T (Fig. 2a).

This indicates now that the process of inside-out activation of integrins is likely to be hampered by the VEYA-mutation and that in addition to the link of talin, an additional mechanical coupling is required to arrest the rearward sliding adhesions. This indicates that the integrin-dependent recruitment of paxillin to focal adhesions, is involved in the maturation of focal adhesions, both with a structural component (LIM domains), as well as with a signaling component (LD motifs).

Reviewer #2

- The manuscript by Marta Ripamonti et al provide a mechanistic insight into the contribution of individual Paxillin LIM domains in targeting and maintaining the structural integrity of FA. While the contribution of paxillin LIM domains acting as scaffolds to target multiprotein complexes to FA sites is known, this work demonstrates that precise spacing of LIM domains within the sequence possibly act as a mechanical tension determinant. Although the idea of proper spacing between LIM domains affecting localization to FAs is previously suggested (Petit, M.M. et al., 2003), the current study has confirmed this experimentally. The authors additionally claim that both LIM1 & 2 are comparable to LIM3 for FAs targeting with LIM2 being the major determinant in contrast to LIM3 as reported earlier (Brown et al 1996). Each LIM1,2,3 have unique functional and spatial signatures and are not interchangeable. Further, authors claim an additional function of Paxillin N-terminus and highlight the importance of tyr31 and 118, phosphorylation sites for FAK and vinculin binding in stabilizing LIM domain dependent paxillin localization at FAs which is in contrary to previous findings (Brown et al., 1996). Additionally, the author highlighted the role of Paxillin LIM4 Cys512 and Cys557 as potential palmitoylation sites that aid in association with PM and integrins and thus influence the maturation of integrin clustering and association with its adaptor proteins. The above-mentioned claims although not entirely novel in conception as most of these findings are inferred from previous studies but has been systematically, experimentally confirmed in this study for the first time using convincing molecular and cellular assays. Overall, this work provides a comprehensive understanding of physiological functioning of FAs and how structural interaction and recruitment of paxillin influence the FAs integrity. To this reviewer the manuscript seems publishable when the authors address the following minor concerns.

We thank the reviewer for this rigorous and fair evaluation of our manuscript, as well as highlighting some important publications that we have not yet cited. Below are point by point responses to the specific issues raised by this Reviewer.

1. Compromised paxillin binding in signaling defective integrin $\beta 3^{VE/YA}$ as demonstrated in (Fig1b, g) is not visually convincing. The representative images show substantial paxillin recruitment and association with $\beta 3^{VE/YA}$. I would suggest doing a colocalization quantification (Mander's coeff.) for paxillin and beta integrin WT & mutants to confirm any substantial difference in paxillin recruitment. The instability of FAs is rather indicative of rapid dissociation of paxillin from $\beta 3^{VE/YA}$ adhesion sites as addressed in the next section rather than compromised paxillin binding.

We thank the reviewer for this comment and the alternative interpretation of the VE/YA mutation. Indeed, our data indicate that this mutation is reducing paxillin recruitment to focal adhesions, but even more so, is not able to physically retain paxillin in these structures. This subtle but relevant difference in interpretation has been added to the text, and is now also consistent with the newly added data to figure 1.

In the first figure of our manuscript we have included the analysis of paxillin localization to $\beta 3$ integrin-positive FAs, which we are claiming to rely on a mechanism distinct from that of binding. Indeed, with the help of the BiFC technique we have demonstrated that even a highly dynamic and FA-binding defective paxillin molecule as the paxillin 1334 is capable of, at least transiently, localize to $\beta 3$ integrin-positive FAs where it could be trapped (Fig. 7i-l). Similarly, the $\beta 3^{VE/YA}$ integrin which contains a mutation of the putative paxillin binding site is not entirely abolishing paxillin recruitment (Fig. 2d,e and Supp. Fig. 3a) but rather proper interaction with this signaling protein, as demonstrated by the

extremely accelerated dissociation rate of PA-GFP_paxillin from $\beta 3^{VE/YA}$ _mCh adhesions (Fig. 3g-i). However, when paxillin is overexpressed, the localization defect is overcome and reflected by the similar paxillin/ $\beta 3$ integrin ratio in the different types of adhesions (Fig. 1d,f). On the contrary, when we only look at the endogenous paxillin, a strong reduction of localization to $\beta 3^{VE/YA}$ integrin-adhesions can be seen (Fig. 1g). The apparently normal recruitment at high paxillin expression levels is strikingly different from the paxillin binding defect, which requires the dynamic analysis for detection (Fig. 3g-i). This means that binding defects can not be determined by a simple analysis of paxillin localization, but needs to be demonstrated by other means, such as the implementation of a dynamic study as our photoactivation experiments.

Accordingly, we cannot agree with the Reviewer's comment: "*Compromised paxillin binding in signaling defective integrin $\beta 3^{VE/YA}$ as demonstrated in (Fig1b, g) is not visually convincing.*" Compromised paxillin binding to the signaling defective integrin $\beta 3^{VE/YA}$ integrin was previously shown specifically during $\beta 3$ integrin-dependent spreading assays in serum free-conditions (Pinon et al. 2014) and now confirmed by means of photoactivation experiments but is not reflected at the level of paxillin "accumulation" at $\beta 3$ integrin-positive adhesions when cells were co-expressing mCherry_paxillin and GFP-tagged integrins and fixed and imaged after an overnight incubation on coverslips with complete medium.

We hope that this point has been made clearer in the new version of the manuscript, in which we showed the mCherry/GFP quantification in adhesions, as well as the localization of endogenous paxillin. Indeed the latter revealed that endogenous paxillin "accumulation" in adhesions is reduced in the presence of the mutant integrins, in particular with the $\beta 3^{VE/YA}$ integrin.

New manuscript, lines 126-129: Interestingly, the overexpressed mCherry_paxillin localized equally well to FAs, independently of the type of $\beta 3$ integrin subunit expressed as a fusion with GFP, as demonstrated by the quantification of its mCherry fluorescence intensity in respect to the GFP signal (Fig. 1d).

Following on the Reviewer's comment: "*The instability of FAs is rather indicative of rapid dissociation of paxillin from $\beta 3^{VE/YA}$ adhesion sites as addressed in the next section rather than compromised paxillin binding.*" A stable protein-protein interaction results in a slow dissociation rate; on the contrary, a compromised binding, in a faster dissociation rate. Being the mutation introduced on the integrin tail and not on paxillin, it could not be the intrinsic dynamic of paxillin that has changed and afterward caused the instability of FAs. In other words, introducing the mutations on the integrin, we modulated the affinity between proteins (as in the case of the VE-dependent stable $\beta 3$ integrin-talin complex), and consequently affected the dynamics of these proteins, not *vice versa*. Most importantly, we have used the dissociation rate of paxillin from $\beta 3$ integrin-positive FAs as a readout of its binding capacity to these sites.

2. Live cell imaging movie for $\beta 3^{VE}$ is missing (line 108). The Supplementary Movies 1 and 2 are for $\beta 3^{WT}$ and $\beta 3^{VE/YA}$.

Live cell imaging movies were initially only provided for $\beta 3^{WT}$ and $\beta 3^{VE/YA}$ integrins, plus in combination of BiFC for the latter. As also suggested by Reviewer #1, the movie of $\beta 3^{VE}$ _GFP adhesions has now been included in the new version of the manuscript (Movie 2), while the others, $\beta 3^{WT}$ and $\beta 3^{VE/YA}$, are now indicated as Movie 1 and 3 respectively.

3. Figure 7d show no significant effect of 2BP treatment on the palmitoylation levels in cell lysates expressing WT mCh Paxillin between control and treated samples. Authors could provide a better representative pulldown image for the Acyl-Resin Capture assay.

[Now corresponding to Fig. 8e].

To address this point we bring to the attention of the Reviewer a few lines of our manuscript (lines 405-409 of the old version or lines 556-560 of the new one).

“The amount of palmitoylated paxillin strongly increased when an irreversible geranylgeranyl lipid anchor was coupled to its carboxyl-terminus by means of the previously exploited CaaX-box (mCherry_paxillin-CAIL). Only in this context we appreciated the effect of the treatment with 2-bromopalmitate (2BP) that, competing with endogenous palmitate, acts as a general inhibitor of palmitate incorporation.”

Given that we observed an effect of the 2BP treatment on the paxillin precipitation *via* the Acyl-RAC only when expressed in the form of mCherry_paxillin-CAIL, it is impossible for us to satisfy this request of Reviewer #2. A possible explanation for the absence of an effect of the 2BP treatment on the palmitoylation of the mCherry_paxillin wt construct could be linked to the low level of paxillin wt palmitoylation, meaning that the fraction of palmitoylated paxillin is much smaller than the total cellular pool of paxillin. According to our results, we postulated that only a small fraction of FA-associated mCherry_paxillin is actually palmitoylated, whereas a much bigger pool of the overexpressed protein is palmitoylated when presenting the geranylgeranyl lipid moiety anchoring it to the plasma membrane. Interestingly, in the latter situation the 2BP-inhibitory effect is clearly visible. Also, for the effective visualization of the 2BP-inhibition of palmitoylation, we would require that none of the cysteines within paxillin is modified by palmitoylation, as only one palmitoylated cysteine is theoretically sufficient for the coupling of the protein to the thiol-reactive beads. We therefore think that in the case of multiple available cysteines, as indicated by the newly added mutagenesis study of the LIM4 domain constructs, the 2BP inhibitory effect would be less pronounced or even not visible. On the other hand, that we can influence the level of palmitoylation of the geranylgeranyl-modified paxillin, clearly indicates that palmitoylation sites exist in paxillin. Interestingly the proteomic analysis of the palmitoylated protein fraction, which we newly included into the manuscript demonstrated a strong overlap with the collection of palmitoylated proteins identified in Swisspalm.

Reviewer #3

- Ripamonti et al., Structural and functional analysis of LIM domain-dependent 1 recruitment of paxillin to focal adhesions

In the present study, Ripamonti et al analyze integrin-dependent mechanisms of the LD and LIM domain-containing protein paxillin recruitment of focal adhesions (FAs) using a set of elegant cell biological and imaging techniques. They show that mechanical coupling of paxillin in the FA to the plasma membrane or integrin is important for FA stability and integrin-talin linkage. Uncoupling paxillin using a mutant $\beta 3$ integrin in a putative paxillin binding site led to increased integrin flow in FAs and reduced mechanical stability of FAs. The authors went on to define the contribution of the individual LIM domains for FA localization and function using a series of paxillin variants (deletion and domain swapping) and demonstrate the spatial orientation of the paxillin LIM domains, with LIM4 closest to the plasma membrane.

In general, the manuscript contains a lot of information, is well written and structured. Although some results are confirmatory the study provides sufficient amount of novel results (contribution of individual LIM domains and the orientation of the paxillin LIM domains) to be of great interest for the community. However, I am not entirely convinced that the experimental setup supports all the conclusions. The imaging techniques including BiFC, the focal adhesion isolation, FA sliding are very elegant but one wonders if some conclusions, e.g. plasma membrane interaction of LIM4, would require additional experiments to draw these conclusions. Also, the presence of endogenous paxillin might mask functions of mutant paxillin variants when co-expressed in the same cell. Additional experiments are necessary to draw the conclusions and to support publication of the study in Communications Biology.

We thank this reviewer for the evaluation of our manuscript and the suggestions of additional experiments that could further strengthen some of our conclusions. In accordance with the comments of the other reviewers we have included some additional data obtained in paxillin-deficient cells, as well as added data to better judge the potential of the LIM4 domain to interact with the plasma membrane.

1. One point of criticism is the expression of proteins in the background of the endogenous proteins. While this might not be problematic in case of the $\beta 3$ integrin variants as NIH-3T3 cells express low levels of endogenous $\beta 3$ integrin, the analysis of paxillin variants could be more problematic. We have observed in the past and studies have shown that GFP-tagged proteins might not be able to localize properly in the presence of the endogenous protein but does so in the absence of the endogenous protein. It would have been a clearer setup to study paxillin in paxillin-depleted NIH-3T3 cells. Although this might be too much for all experiments it should be mentioned in the results of discussion section.

According to the comments of Reviewers #1 and #3, we have tried to better determine the role of the endogenous paxillin in our experiments and to understand if it would influence the outcome and/or the interpretation of our results. While it wasn't feasible for us to generate a paxillin-null cell line starting from the clone of NIH-3T3 cells that we have been using so far, we have extended our analysis to murine paxillin-deficient fibroblasts, kindly provided by Prof. Daniel Bouvard (Centre de Recherche en Biologie cellulaire de Montpellier). In these paxillin-null cells, we tested the dissociation kinetics of photoactivated paxillin constructs (Supp. Fig. 7a-c), as well as evaluated the FA-localization of wild-type and mutant versions of overexpressed paxillin (Supp. Fig. 7d-g). Interestingly, the recruitment of overexpressed paxillin variants to transfected $\beta 3^{WT}$ _GFP integrin was not different from that observed in NIH-3T3 fibroblasts (Supp. Fig. 7d-g). Although some of the most deleterious paxillin mutations

reduced the accumulation to FAs (Supp. Fig. 7d-i), paxillin localization to FAs was never completely prevented. We are aware that such residual recruitment to focal adhesions appears to be the result of paxillin overexpression, but in light of our photoactivation experiments, where mutant forms of paxillin were rapidly dissociating, would indicate that indeed, low and high affinity interactions are controlling the recruitment to FAs, similar to what has been previously proposed by Wolfenson and coworkers (Wolfenson et al. 2009).

Also, the phenotype in terms of adhesions size, length and number, was similar between $\beta 3$ integrin-positive FAs in NIH-3T3 and paxillin-null fibroblasts (Fig. 1h-m). While it became clear that differences in paxillin recruitment to mutated $\beta 3$ integrins was visible at the level of endogenous paxillin detected by antibody staining, such differences were reduced when paxillin was overexpressed, either in the presence or in the absence of endogenous paxillin (Fig. 1d,f). In addition, we were also able to exclude a differential contribution of endogenous $\beta 1$ integrins to the recruitment of paxillin to wild-type or mutant forms of $\beta 3$ integrin-containing FAs (Supp. Fig. 1a,b).

According to the data gathered from the newly included experiments, we do expect a competition between the exogenously expressed paxillin and the endogenous isoform. While we cannot completely exclude that this can influence the localization and the dynamics of even the fluorescently-tagged wild-type full-length paxillin, we also caution the result of experiments produced in a paxillin-deficient background. In fact, the paxillin's closely related family member Hic-5 was shown to have, to some extent, overlapping functions with paxillin (Deakin and Turner 2011). In addition, it was shown that Hic-5 could not be deleted in paxillin-deficient cell without affecting cell spreading and FAK-phosphorylation, suggesting that in the absence of paxillin, Hic-5 takes over some paxillin specific tasks (Pasapera et al. 2010). It is therefore likely that in the absence of paxillin, as in the paxillin-null fibroblasts that we have used, the exogenously expressed paxillin molecules compete with the endogenous Hic-5. While in a context of paxillin-positive cells we can easily identify the differences between the exogenous proteins introduced and the competing endogenous paxillin (e.g. presence of the tag, mutations, deletions), the analysis would be even more complex if we need to consider the competition with, and the function of, the endogenous Hic-5 compensating for the absence of paxillin.

It was however of great interest to performed some key experiments in paxillin-null cells and validate our findings about paxillin in the NIH-3T3 cell line (Supp. Fig. 7a-c). This issue is also addressed in point 1 raised by Reviewer #1.

2. What is the effect of a paxillin knockout or knockdown on the FAs sliding phenotype, FA stability etc? Would the authors expect a similar phenotype to the $\beta 3^{VE/YA}$ integrin? This would illustrate how much of paxillin function in FAs is attributed to its localization within FAs and to its proper mechanical linkage to the membrane/integrins.

We do not expect that a KO/KD of paxillin would generate a phenotype similar to the sliding of the $\beta 3^{VE/YA}$ integrin. It was previously shown in paxillin deficient ES cells that cell spreading and FAK-phosphorylation was affected by the complete removal of paxillin-family proteins. Because of the absence of cell spreading in the first place, it would be very difficult to investigate integrin-dependent adhesions and signaling. Although we may not be able to recapitulate this phenotype with a paxillin mutant, reconstitution experiments of paxillin fragments have shown that either LIM domain-deletion mutants, or the absence of LD motifs are affecting FAK-phosphorylation and cell spreading (Wade and Vande Pol 2006). In our hands, physical tethering of either a LIM domains-only construct, or only the paxillin N-terminus, to $\beta 3^{VE/YA}$ integrin is able to rescue the sliding phenotype, proposing that both the

enhanced physical stability as well as the recruitment of signaling molecules to LD motifs are critical in inducing a stabilization of the integrin anchorage.

The defects of stability of the $\beta 3^{VE/YA}$ integrin could be related to concomitant phenomena: the loss of paxillin binding, an inefficient inside-out activation mechanism of the receptor and the stronger talin binding, which would strengthen the link with the actin cytoskeleton. Following our data on the previously published activation state of $\beta 3^{WT}$, $\beta 3^{VE}$, $\beta 3^{VE/YA}$ and $\beta 3^{YA}$ integrins corresponding to 1, 2, 1 and 0.5 respectively (Pinon et al. 2014), we have investigated whether a firm interaction of the $\beta 3^{VE/YA}$ integrin and the ECM (obtained by the N³⁰⁵T mutation) would have arrested the sliding phenotype. In light of this rescue and the correlation between the actin and the $\beta 3^{VE/YA}$ integrin flows, we expect that the reduced ECM interaction, plus the stronger actin connection *via* the high-affinity talin-binding site, would be the mechanism underlying the rapid inward sliding. This means that at the origin of the defective sliding phenotype is the impaired inside-out activation of the $\beta 3^{VE/YA}$ integrin which prevented a robust integrin/ECM linkage. In fact, recent data from our lab demonstrate that talin binding to the integrin tail is not directly correlating with the F1-loop-induced activation of the integrin receptor. Therefore it appears that the $\beta 3^{VE/YA}$ integrin mutant is able to physically interact with talin, but that regulatory mechanisms are missing to maintain the integrin receptor in a fully activated state. This latter hypotheses is now also supported by the impossibility to retain $\beta 3^{VE/YA}$ integrin during the FA-isolation procedure, unless presenting the N³⁰⁵T substitution or the mechanically connected paxillin (Fig. 4g,h), corresponding to new data that we have now introduced into the manuscript.

3. The authors rescued the $\beta 3^{VE/YA}$ integrin the FA sliding phenotype by BiFC-based mechanical coupling of paxillin to $\beta 3^{VE/YA}$ integrin and suggest that a compromised paxillin binding and retention within FAs is the underlying cause of the sliding phenotype. Was it analyzed if the BiFC-based method indeed led to increased mechanical coupling of paxillin to $\beta 3^{VE/YA}$ integrin e.g. upon isolation of FA complexes (done in Figure 3). Did the authors examine the expression levels of mCherry-paxillin vs mCherry-paxillin-CN (or CC) to rule out that dissimilar protein levels cause the observed effects?

We thank the reviewer for these excellent questions, revealing in more details how the mechanical coupling of paxillin to the $\beta 3^{VE/YA}$ integrin is linked to the stability of focal adhesion complexes. We have therefore included additional data in the new version of our manuscript (Fig. 4g,h), which tested the outcome of the expression of paxillin tagged with the BiFC probe or of its the mechanical coupling to integrin *via* BiFC.

In terms of FA-isolation, the BiFC-dependent mechanical coupling of paxillin to the $\beta 3^{VE/YA}$ integrin made possible the identification of properly extracted and isolated mCherry_paxillin_CC and $\beta 3$ integrin-positive adhesions (Fig. 4h). Additionally, we ruled out the possibility that dissimilar protein levels between mCherry_paxillin and mCherry_paxillin_CC are the cause of the differential rescuing by analyzing $\beta 3^{VE/YA}$ _GFP flow in the presence of mCherry_paxillin_CC (Fig. 2h, ctr). To summarize, we previously included two conditions:

- $\beta 3^{VE/YA}$ _GFP + mCherry_paxillin [rapid sliding]
- $\beta 3^{VE/YA}$ _CN + mCherry_paxillin_CC [rescue]

We now added:

- $\beta 3^{VE/YA}$ _GFP + mCherry_paxillin_CC [rapid sliding]

mCherry_paxillin_CC did not rescue the sliding of $\beta 3^{VE/YA}$ integrin, unless the latter presented the complementary citrine fragment.

4. How do the authors explain the reduced half-life of paxillin in $\beta 3^{VE}$ integrin adhesions despite comparable integrin flow in adhesions? Isn't this contradicting the authors hypothesis that paxillin stability in FAs and its coupling to integrins is the underlying cause of the sliding phenotype?

We did detect a faster dissociation of paxillin from $\beta 3^{VE}$ integrin-adhesions than from the wt ones. However, this rate of dissociation was not comparable to that observed in $\beta 3^{VE/YA}$ integrin adhesions, meaning that there could be a certain stability threshold which is nevertheless reached in the $\beta 3^{VE}$ integrin-adhesions, allowing the following stabilization of the integrin/talin complex. Additionally, we have now added experiments suggesting that the rapid sliding phenotype is the result of a series of co-occurring events (see above). If the same reasoning is applied to the $\beta 3^{VE}$ integrin-adhesions, we would not expect its sliding because of its enhanced activation state, compared to that of the wt and $\beta 3^{VE/YA}$ integrin, which allows a firm interaction with the ECM.

5. Figure 4: It is surprising that LIM-only rescue the adhesion flow while the N-terminal part with the two tyrosine residues contribute the paxillin stability in FAs (figure 4). Did the authors use the paxillin N-terminal half for the BiFC experiments in figure 4i or any other unrelated citrine-tagged protein as negative control to rule out that any citrine-tagged protein would slow down the flow in adhesions?

[Now corresponding to Fig. 5]

Following on this excellent point raised by Reviewer #3 we have performed the following experiments to measure the $\beta 3^{VE/YA}$ integrin flow in adhesions, when mechanically coupled *via* BiFC with:

- mCherry_CN (Fig. 2h)
- mCherry_paxN-term_CC (Fig. 5i)

While the BiFC established with the mCherry protein expressed only fused to the citrine fragment did not rescue the sliding phenotype (Fig. 2h), the presence of the paxillin N-terminus did slow the focal adhesion sliding to levels comparable to the mechanical coupling of LIM domains (Fig. 5i). As discussed above we think that this results proposes that indeed either the tethering of some LD motif-interacting proteins could induce the adhesion signaling required for focal adhesion maturation, or that the mechanical stabilization, and potentially talin or kindlin-mediated signaling could also lead to focal adhesion stabilization. Further experiments are however required to dissect these mechanisms in more detail.

6. Figure 6,7: Did any of the mutant (LIM domain replacement mutants or point mutant is the LIM4 domain) affect localization of the respective paxillin construct to FAs? Perhaps the authors could include the TIRF images of the different variants as supplementary figure.

All the tested paxillin mutants did localize to $\beta 3$ integrin-positive adhesions, both in the case of LIM4 mutations as well as in the case of the LIM domain replacements. We have now included TIRF images of some of these paxillin mutants co-expressed with $\beta 3^{WT}$ _GFP (Fig. 7b and Fig. 8c). Concerning the mechanisms of the low-affinity recruitment of the FA-binding deficient paxillin mutants, we have not yet identified a specific cause. However, we included data demonstrating a LIM domains-dependent interaction of paxillin with the head-domain of talin, which adds to the possible interaction partners contacted by paxillins during its engagement with focal adhesions. Nevertheless, it is important to mention that the recruitment of paxillin to the integrin complex requires the application of tension. This also means that relevant paxillin interaction sites are masked under low tension, or specifically revealed if structural elements of focal adhesions, such as F-actin fibers are exposed to stress as proposed in a recent paper ("Mechanosensing through Direct Binding of Tensed F-Actin by LIM

Domains” now added to the citation list). Importantly, our study provides an experimental toolbox to study with high precision the mechanisms and contribution of the different LIM domains to the recruitment and subsequent binding of paxillin to focal adhesions.

7. Figure 7: The authors write that they used “site-directed mutagenesis, preventing palmitoylation at Cys512/557 or abrogating LIM4 positive charge, did not affect paxillin expression levels” (line 391f). However, they did not test if the cysteine mutations indeed affect palmitoylation. This should be included and would be a perfect control to show that their Acyl-Resin Assisted Capture assay indeed shows a small but specific pool of palmitoylated paxillin. (the authors do write in the discussion that one would need to screen 24 cysteines but since they only write about palmitoylation in the context of the LIM4 domain the results part, it should be tested).

When performing Cys-to-Ser mutations in order to prevent palmitoylation at a given site, it is in fact very difficult to assure that palmitoylation will not occur at another site in the same protein, especially when the initial membrane recruitment mechanisms are still functional, and other potential palmitoylation sites are nearby. Unfortunately, the palmitoylation levels of the endogenous and transfected paxillin proteins are clearly inferior to our positive control, caveolin. In addition to Cys⁵¹² and Cys⁵⁵⁷ of LIM4, each LIM domain has 5 to 6 cysteines involved in complexing the Zn²⁺ ions. Although the literature is sparse in respect to such Zn²⁺ complexed Cys-residues and whether they would potentially be available for palmitoylation, the second zinc-finger in the LIM4 domain is particular, as it is the only one of the paxillin LIM domains consisting of 4 cysteines complexing the single Zn²⁺ ion. Despite this particular feature, we have so far not attempted to mutate the Zn²⁺ coordinating cysteines, in order not to jeopardize the structure of the LIM domain.

Paxillin has 25 cysteines, among which 22 are involved in the binding of the zinc atoms of LIM domains. The remaining three corresponds to Cys¹⁰⁸ (pax N-term) and Cys⁵¹² and Cys⁵⁵⁷ (within LIM4). While the initial thinking was that cysteines involved in zinc-binding couldn't, at the same time, be palmitoylated, the acyl-RAC experiments with the triple Cys-to-Ser paxillin mutant showed that indeed this is possible (data not shown in the manuscript). In fact, the palmitoylation of paxillin^{C108/512/557S} must have occurred on at least one cysteine coordinating the binding of zinc atoms within the LIM domains. However, it is also possible to speculate based on our functional data that Cys⁵¹² is the major palmitoylation site, leading to a strong reduction in FA-retention. In its absence however, and when combining the mutation of the cysteine at position 557, at least one alternative site needs to be palmitoylated to allow precipitation in the acyl-RAC assay. Noteworthy, the complexity of the experimental procedure made difficult to quantitatively judge the level of S-acylation among the recombinant proteins, and the only certain evaluation was that the geranylgeranyl-anchor boosted the palmitoylation as well as rescued the reduction in FA-binding upon mutation of Cys^{512/557}. Moreover, if the presence of a single palmitoylated cysteine is sufficient for protein coupling to thiopropyl sepharose beads, then potentially all the palmitoylated paxillin molecules can be trapped with the same efficiency, independently of the number of modified (acylated) cysteines. We couldn't therefore proof, nor reject, the hypothesis that paxillin palmitoylation occurs on Cys 512 and/or 557.

Nevertheless and in order to convince the readers that our experimental strategy was efficient, we included a proteomic analysis of the potentially palmitoylated proteins, pull-down during the Acyl-RAC assay, which allowed the detection of the transfected paxillin protein, as well as 200 proteins, among which more than 90% had been previously annotated and listed as palmitoylated proteins in the Swisspalm database (new figure 8g,h).

8. Figure 8: The mutational analysis of paxillin would benefit if some of the mutant e.g. Cys512/557 or abrogating LIM4 positive charge would be included in the BiFC analysis (Figure 8b,d) to determine if any of these mutations indeed affect the proper orientation of paxillin to the plasma membrane or $\beta 3$ integrin.

[Now Fig. 9]

We had already asked ourselves the same question and undertaken some of the suggested BiFC experiments. In particular, BiFC between *paxillin* Δ LIM4 or *paxillin* LIM4 triple mutant (RKK-QED) and the PM were tested (however in NIH-3T3 cells without co-expression of $\beta 3$ integrin). None of them prevented or strongly impaired the BiFC efficiency between the paxillin C-terminus and the PM-localized probed.

While frustrating in the beginning, we have then considered the limited consequences of LIM4 deletion/mutations, when compared for example to the other LIM domain deletions. Although possibly preventing a direct interaction with the PM, the deletion/mutation of LIM4 would not prevent the correct orientation of paxillin, thus the mutated LIM4 domains may still be positioned in the proximity of the plasma membrane. The emerging complex network of interactions of the paxillin LIM domains would suggest that when abolishing one/some of these interactions, the remaining will locate paxillin in the best possible position to engage as many binding partners as possible. The deletion of LIM4 did not preclude the interaction of the LIM1-LIM2-LIM3 array with the respective interactors in FAs, which would therefore orient correctly the protein, so that LIM3 would now be the closest to the PM and still sufficiently in the proximity for establishing BiFC (probes located in the PM and at the C-terminus of mCherry_paxillin Δ LIM4).

We have therefore reformulated some of our findings for example correcting the overstatement that *LIM4 domain orients the paxillin protein*, while it *could stabilize paxillin orientation*, possibly interacting with the PM.

Minor points

1. It would be beneficial for readers if a brief description of the NIH-3T3 cells could be included; specifically, that these cells express low levels of $\beta 3$ integrin endogenously (this is mentioned in a previous publication by the Wehrle-Haller lab but could be mentioned here again). While this explains that the phenotype of transiently-expressed mutant $\beta 3$ integrins is prominent I was puzzled by the fact that $\beta 1$ integrins do not seem to play a major role for mCherry-paxillin localization (no paxillin in $\beta 1$ integrin containing fibrillar adhesion) or FA displacement despite the fact that cells were grown in full medium (which should contain FN and VN) and without special coating of surfaces with VN.

In light of the Reviewers' comments we have included a better description of the NIH-3T3 cells and a broader analysis addressing the expression of $\beta 3$ _GFP integrins and the activation of $\beta 1$ integrins. The latter integrin appeared to only marginally contribute to the formation of focal and fibrillar adhesions in $\beta 3$ integrin-transfected NIH-3T3 and paxillin-null fibroblasts, when grown in serum containing conditions. However, in other cell types and/or using different culturing conditions (e.g. lower serum contents) activated $\beta 1$ integrin may become a major receptor of adhesions (Supp. Fig. 1).

It is not surprising to see that in the chosen culture conditions, the reactivity of the 9EG7 mAb was limited when used to stain NIH-3T3 fibroblasts cultured in complete medium for an overnight period. On the other hand, when studying $\beta 1$ integrins we do require a fibronectin coating and/or longer culturing times, to see efficient fibrillar adhesion formation. In fact, due to the different abundance of VN and FN in FBS, VN was identified as the main adhesive protein in routine cell culture media (Hayman et al. 1985). According to the amount of substrate bound adhesion proteins, when we plated

the same cells on fibronectin-coated glass coverslips, a nice pattern of activated $\beta 1$ integrins was detected in focal as well as centrally located adhesions, proposing that on FN-coated glass $\alpha 5\beta 1$ is likely to be dominant, while on serum derived VN-coated glass $\alpha v\beta 3$ integrin is the dominant integrin found in FAs (Soto-Ribeiro et al. 2019).

2. If possible, the authors should quantify the FA instability and accelerated retrograde sliding shown in the supplementary movies 1+2.

The specific integrin sliding measured for the movies 1, 2 and 3 is now provided in Fig. 1n, while the result of the sliding analysis of all experimental movies is provided in Fig. 2a as a box plot.

3. Line 213f: The authors write “Indeed, while in intact cells, paxillin was recruited to $\beta 3$ WT and $\beta 3$ VE adhesions equally well (Fig. 1b), in isolated FAs, the...”. However, the paxillin recruitment to $\beta 3$ WT and $\beta 3$ VE adhesions was not quantified and only analyzed qualitatively. The authors would need to measure fluorescence intensities and determine their ratios ($\beta 3$ vs paxillin) to make such a statement.

Quantification is now provided in Fig. 1d for co-expressed paxillin and $\beta 3$ integrin in NIH-3T3 and in Fig. 1f for their co-expression in paxillin-null cells.

4. Please include the reference for the Acyl-Resin Assisted Capture assay. Gadalla, Abrami and Goot is written in the methods section but not listed in the references.

The reference has been properly included in the reference section.

- Deakin, N. O., and C. E. Turner. 2011. 'Distinct roles for paxillin and Hic-5 in regulating breast cancer cell morphology, invasion, and metastasis', *Mol Biol Cell*, 22: 327-41.
- Hayman, E. G., M. D. Pierschbacher, S. Suzuki, and E. Ruoslahti. 1985. 'Vitronectin--a major cell attachment-promoting protein in fetal bovine serum', *Exp Cell Res*, 160: 245-58.
- Pasapera, A. M., I. C. Schneider, E. Rericha, D. D. Schlaepfer, and C. M. Waterman. 2010. 'Myosin II activity regulates vinculin recruitment to focal adhesions through FAK-mediated paxillin phosphorylation', *J Cell Biol*, 188: 877-90.
- Pinon, P., J. Parssinen, P. Vazquez, M. Bachmann, R. Rahikainen, M. C. Jacquier, L. Azizi, J. A. Maatta, M. Bastmeyer, V. P. Hytonen, and B. Wehrle-Haller. 2014. 'Talin-bound NPLY motif recruits integrin-signaling adapters to regulate cell spreading and mechanosensing', *J Cell Biol*, 205: 265-81.
- Soto-Ribeiro, M., B. Kastberger, M. Bachmann, L. Azizi, K. Fouad, M. C. Jacquier, D. Boettiger, D. Bouvard, M. Bastmeyer, V. P. Hytonen, and B. Wehrle-Haller. 2019. 'beta1D integrin splice variant stabilizes integrin dynamics and reduces integrin signaling by limiting paxillin recruitment', *J Cell Sci*, 132.
- Thievensen, I., P. M. Thompson, S. Berlemont, K. M. Plevock, S. V. Plotnikov, A. Zemljic-Harpf, R. S. Ross, M. W. Davidson, G. Danuser, S. L. Campbell, and C. M. Waterman. 2013. 'Vinculin-actin interaction couples actin retrograde flow to focal adhesions, but is dispensable for focal adhesion growth', *J Cell Biol*, 202: 163-77.
- Wade, R., and S. Vande Pol. 2006. 'Minimal features of paxillin that are required for the tyrosine phosphorylation of focal adhesion kinase', *Biochem J*, 393: 565-73.
- Wolfenson, H., A. Lubelski, T. Regev, J. Klafter, Y. I. Henis, and B. Geiger. 2009. 'A role for the juxtamembrane cytoplasm in the molecular dynamics of focal adhesions', *PLoS One*, 4: e4304.
- Zhang, P., L. Azizi, S. Kukkurainen, T. Gao, M. Baikoghli, M. C. Jacquier, Y. Sun, J. A. E. Maatta, R. H. Cheng, B. Wehrle-Haller, V. P. Hytonen, and J. Wu. 2020. 'Crystal structure of the FERM-folded talin head reveals the determinants for integrin binding', *Proc Natl Acad Sci U S A*.

REVIEWERS' COMMENTS:

Reviewer #1 (Remarks to the Author):

The authors have provided a detailed response and the revised manuscript addresses satisfactorily my major concerns through the inclusion of additional data and editorial modifications.

Importantly, they have performed some basic additional experiments in a pax null background and documented results accordingly in the text. Nevertheless, it remains unclear to me as to why they chose to do these analyses in a completely new CRISPR pxn KO line rather than perform transient pxn KD in their b3 WT and mutant stable NIH 3T3 cell lines.

Indeed, their rationale regarding whether or not to perform these KO/KD experiments at all is also somewhat confusing. (See responses to Rev 1, point 1 and Rev 3, points 1 and 2). Their assertion, in response to Rev 1 point 1, that depletion of pxn would result in the failure to form any focal adhesions, or even prevent cell spreading (response to Rev 3, point 2) is clearly not supported by their own data or that of numerous previous publications by others and therefore these comments aren't really relevant.

Despite these apparent contradictions in the rebuttal, this is a very thorough study and I support publication without further revisions.

Reviewer #2 (Remarks to the Author):

The queries raised during the 1st review are now being addressed. The reiteration of manuscript text in most part have provided clarity in the interpretation of the results. Along with newly added data and by providing the missing datasets as have been asked for earlier, all the raised concerns have been addressed.

I would hence recommend the manuscript for publication with no further revisions.

Reviewer #3 (Remarks to the Author):

The authors addressed all my concerns and after a careful analysis of the revised manuscript I can support publication in Communications Biology.

Three minor points:

Figure 2a: How does the N305 mutation behave on its own? If the authors have the data at hand they should consider including it. However, I would not insist on additional experiments to address this point.

Figure 7b and Supplementary Figure 2d: Include scale bar.

Line 366: I would write about "adhesion complexes"

REVIEWERS' COMMENTS:

Reviewer #1 (Remarks to the Author):

The authors have provided a detailed response and the revised manuscript addresses satisfactorily my major concerns through the inclusion of additional data and editorial modifications.

Importantly, they have performed some basic additional experiments in a pax null background and documented results accordingly in the text. Nevertheless, it remains unclear to me as to why they chose to do these analyses in a completely new CRSPR pxn KO line rather than perform transient pxn KD in their b3 WT and mutant stable NIH 3T3 cell lines.

Indeed, their rationale regarding whether or not to perform these KO/KD experiments at all is also somewhat confusing. (See responses to Rev 1, point 1 and Rev 3, points 1 and 2). Their assertion, in response to Rev 1 point 1, that depletion of pxn would result in the failure to form any focal adhesions, or even prevent cell spreading (response to Rev 3, point 2) is clearly not supported by their own data or that of numerous previous publications by others and therefore these comments aren't really relevant.

Despite these apparent contradictions in the rebuttal, this is a very thorough study and I support publication without further revisions.

Answer

We thank the reviewer for the positive evaluation of our revised manuscript and importantly for having provided critical inputs that have helped to expand and consolidate our findings.

We also agree that the expected phenotypes of paxillin-deleted or paxillin knockdown cells depends on the cellular context this deletion occurs. In many cultured cell lines, the deletion of paxillin is compensated by the expression of Hic-5 (Petropoulos et al. 2016). It was reported that either paxillin only or Hic-5 only cells behave similar in 2D cultures, but show differences when cultivated in a 3D matrix (Deakin and Turner 2011). When paxillin is knocked out in cells that do not express Hic-5, a lack of cell spreading has been observed (Pasapera et al. 2010; Wade, Bohl, and Vande Pol 2002). In this particular situation, expressing only the LIM domains did not rescue the spreading defect. Although the latter case appears to be an experimentally cleaner set-up. Analyzing the role of individual LIM-domains as performed in our study would not be possible.

Reviewer #2 (Remarks to the Author):

The queries raised during the 1st review are now being addressed. The reiteration of manuscript text in most part have provided clarity in the interpretation of the results. Along with newly added data and by providing the missing datasets as have been asked for earlier, all the raised concerns have been addressed.

I would hence recommend the manuscript for publication with no further revisions.

Answer

We thank the reviewer for his or her suggestions, which we have used to enhance the clarity of our manuscript.

Reviewer #3 (Remarks to the Author):

The authors addressed all my concerns and after a careful analysis of the revised manuscript I can support publication in Communications Biology.

Three minor points:

Figure 2a: How does the N305 mutation behave on its own? If the authors have the data at hand they should consider including it. However, I would not insist on additional experiments to address this point.

Figure 7b and Supplementary Figure 2d: Include scale bar.

Line 366: I would write about “adhesion complexes”

Answer

We thank the reviewer for this attentive evaluation of our manuscript and his positive feedback.

We have not analyzed the sliding of the $\beta 3^{N305T}$ integrin mutant in this study, however previous experiment from our lab, with similar time lapse acquisitions, have shown that these adhesions are stable in time and position (Cluzel et al. 2005). In addition to these old and long published FRAP analysis, we have performed more recent studies, confirming the reduced exchange rate of this mutant compared to $\beta 3^{WT}$ integrin (Bachmann et al. 2020). Already at the time, we were puzzled by the observation that an integrin activation mutation, occurring at the inner membrane clasp (D⁷²³A) induced enhanced integrin clustering, but no increase in the integrin exchange rate ((Cluzel et al. 2005). How paxillin could be involved in reducing the integrin dynamics needs to be shown in the future, but we have also previously reported that the expression of a dominant active form of Rac1 stabilized the integrin exchange, in a level comparable to the N³⁰⁵T mutation. It is therefore likely that the paxillin recruitment to talin/kindlin activated integrins, is inducing a structural reinforcement of the ECM/integrin/actin linkage.

Scale bars have been included where missing and the text in line 366 modified as suggested.

- Bachmann, M., M. Schafer, V. V. Mykuliak, M. Ripamonti, L. Heiser, K. Weissenbruch, S. Krubel, C. M. Franz, V. P. Hytonen, B. Wehrle-Haller, and M. Bastmeyer. 2020. 'Induction of ligand promiscuity of alphaVbeta3 integrin by mechanical force', *J Cell Sci*, 133.
- Cluzel, C., F. Saltel, J. Lussi, F. Paulhe, B. A. Imhof, and B. Wehrle-Haller. 2005. 'The mechanisms and dynamics of (alpha)v(beta)3 integrin clustering in living cells', *J Cell Biol*, 171: 383-92.
- Deakin, N. O., and C. E. Turner. 2011. 'Distinct roles for paxillin and Hic-5 in regulating breast cancer cell morphology, invasion, and metastasis', *Mol Biol Cell*, 22: 327-41.
- Pasapera, A. M., I. C. Schneider, E. Rericha, D. D. Schlaepfer, and C. M. Waterman. 2010. 'Myosin II activity regulates vinculin recruitment to focal adhesions through FAK-mediated paxillin phosphorylation', *J Cell Biol*, 188: 877-90.
- Petropoulos, C., C. Oddou, A. Emadali, E. Hiriart-Bryant, C. Boyault, E. Faurobert, S. Vande Pol, J. R. Kim-Kaneyama, A. Kraut, Y. Coute, M. Block, C. Albiges-Rizo, and O. Destaing. 2016. 'Roles of paxillin family members in adhesion and ECM degradation coupling at invadosomes', *J Cell Biol*, 213: 585-99.
- Wade, R., J. Bohl, and S. Vande Pol. 2002. 'Paxillin null embryonic stem cells are impaired in cell spreading and tyrosine phosphorylation of focal adhesion kinase', *Oncogene*, 21: 96-107.